# Length independent generalization bounds for deep SSM architectures via Rademacher contraction and stability constraints

**Dániel Rácz**                                      *racz.daniel@sztaki.hun-ren.hu*
*HUN-REN SZTAKI and ELTE, Budapest, Hungary*

**Mihály Petreczky**                                 *mihaly.petreczky@centralelille.fr*
*Univ. Lille, CNRS, Centrale Lille, UMR 9189 CRIStAL, Lille, France*

**Bálint Daróczy**                                   *daroczy.balint@sztaki.hun-ren.hu*
*HUN-REN SZTAKI, Budapest, Hungary*

**Reviewed on OpenReview:** *https://openreview.net/forum?id=Vo6wHBv07k*

## Abstract

Deep SSM models like S4, S5, and LRU are made of sequential blocks that combine State-Space Model (SSM) layers with neural networks, achieving excellent performance on learning representations of long-range sequences. In this paper we provide a PAC bound on the generalization error of non-selective architectures with *stable* SSM blocks, that does not depend on the length of the input sequence. Imposing stability of the SSM blocks is a standard practice in the literature, and it is known to help performance. Our results provide a theoretical justification for the use of stable SSM blocks as the proposed PAC bound decreases as the degree of stability of the SSM blocks increases.

## 1 Introduction

The challenge of learning rich representations for long-range sequences (time series, text, video) has persisted for decades. RNNs, including LSTMs (Hochreiter, 1997) and GRUs (Cho, 2014), struggled with long-term dependencies, while Transformers, despite improvements, still perform poorly on difficult tasks (Huang et al., 2024; Amos et al., 2024).

Recently, several novel architectures have been proposed which outperform previous models by a significant margin, for an overview see Huang et al. (2024); Amos et al. (2024). One notable class of such architectures are the so-called deep State-Space Models (deep SSMs), which typically contain several layers made of the composition of dynamical systems of either continuous or discrete time, and non-linear transformations (e.g. Multilayer Perceptron (MLP), defined in Definition 4.4) (Gu et al., 2021b; 2023; Gu & Dao, 2023; Wang et al., 2024; Gu et al., 2021a; 2022; Smith et al., 2022; Fu et al., 2022; Orvieto et al., 2023). While SSM architectures have been extensively validated empirically, the theoretical foundations of SSMs are less understood. One key point of these models is that they are - often implicitly - equipped with some form of stability constraints for the SSM components. This motivates the question:

> *What is the role of stability in the success of deep SSM architectures for long-range sequences?*

We partially address this problem by leveraging stability to derive a PAC bound which is independent of input sequence length. Our contributions are:

**Stability related system norms** for bounding the Rademacher complexity. We show that the Rademacher complexity of SSMs can be upper bounded by their system norms, such as the $H_2$ and $\ell_1$ norms (Chellaboina et al., 1999), which are well-known in control theory and linked to quadratic stability. This highlights stability

not just as a practical necessity, but as a fundamental aspect of SSM architectures, making it the key takeaway of our work.

**Rademacher Contractions.** We upper bound the Rademacher complexity of multilayer deep SSM models, encompassing many popular architectures, by introducing the concept of *Rademacher Contraction* (RC), which, similarly to the celebrated Talagrand's Contraction Lemma (Ledoux & Talagrand, 1991), allows us to directly bound the Rademacher complexity of deep models that employ nonlinearities. However, in contrast to Talagrand's contraction lemma which requires these nonlinearities to be fixed, i.e., to be independent of the model parameters, RC also handles the case when these nonlinearities are parametrised. Instead, we use a weaker assumption regarding bounded system norms that measure the degree of stability of dynamical systems and are widely used in control theory. In other words, the concept of Rademacher Contractions links a handy tool from the control literature to machine learning.

**PAC bound for stable deep SSMs.** Using the concept of *Rademacher Contraction* we establish a PAC-bound on the generalization error of deep SSMs. The resulting bound is independent of the input sequence length due to stability, and depends only implicitly on the depth of the model. Our results cover both classification and regression tasks for most of the popular deep SSM architectures.

**Outline of the paper.** In Section 2 we present the related literature, then we set some notation and present an informal statement of our result in Section 3. The formal problem statement along with the remaining notation and our assumptions are in Section 4. We propose our main result and a sketch of the proof in Section 5. A numerical example illustrating the result is in Section 6. The majority of the proofs and some additional details are shown in the Appendix.

## 2 Related work

Apart from Liu & Li (2024); Nishikawa & Suzuki (2024), SSM research primarily addresses modeling power, parametrization, and computational complexity, with limited focus on generalization bounds.

**Theoretical analysis of SSMs.** SSM modeling power has been studied via approximation capabilities (Cirone et al., 2024; Wang & Li, 2024; Orvieto et al., 2024), with a survey in Tiezzi et al. (2024). This paper, however, focuses on statistical generalization bounds. Nishikawa & Suzuki (2024) derives statistical bounds on SSM approximation error, but only for a specific learning algorithm and parametrization, whereas our PAC bound is algorithm-agnostic. Experimental results suggest stable SSM parametrizations improve learning (Wang & Li, 2024; Parnichkun et al., 2024; Gu et al., 2021a; Smith et al., 2022; Fu et al., 2022; Gu & Dao, 2023; Smékal et al., 2024). Computational complexity of inference and learning has been analyzed in Massaroli et al. (2023); Gu et al. (2021b; 2023), while Gu et al. (2023); Yu et al. (2024); Wang & Li (2024) investigated initialization techniques.

**PAC bounds for single-layer SSMs.** Liu & Li (2024) derived a PAC bound for a single continuous-time LTI SSM using Rademacher complexity. In this context, LTI refers to Linear Time-Invariant systems and is defined in discrete time in equation 1. In contrast, their result applies only to a single SSM block without nonlinear elements, their bound grows with sequence length, and it does not account for discretization effects. Moreover, their constants are not directly linked to control-theoretic quantities like $H_2/\ell_1$ norms. Their input assumptions also differ: while we assume bounded $\ell_2$ norm inputs, they allow unbounded inputs, but require subgaussianity and continuity, the latter being inapplicable in discrete time and potentially imposing constraints on the sampling mechanism.

**PAC bounds for RNNs.** Since LTIs are core components of deep SSMs and a subclass of RNNs, RNN generalization bounds are somewhat relevant. Note however, that deep SSMs and simple RNNs are theoretically different models and we do not see any trivial way to formulate a deep SSM as a simple RNN or vice verse, see Remark 4.9. Prior PAC bounds for RNNs use VC-dimension or covering numbers (Koiran & Sontag, 1998; Sontag, 1998; Hanson et al., 2021b), Rademacher complexity (Wei & Ma, 2019; Akpinar et al., 2020; Joukovsky et al., 2021; Chen et al., 2020; Tu et al., 2020), or PAC-Bayesian methods (Zhang et al., 2018). However, prior PAC bounds grow with integration time (continuous) or time steps (discrete), limiting their use for long-range sequences. The bound for single vanilla RNNs in Chen et al. (2020) assumes that the state matrix represents a contraction and upper-bounds the $\ell_1$ norm used here. In contrast, our

bound is independent of state-space dimension, assumes only that the state matrix is Schur and assumes only bounded $\ell_1/H_2$ norms. The work Mitarchuk et al. (2024) also assumes that the state matrix is a contraction, works with tanh activation and prove a PAC-Bayesian bound for RNNs under a special saturation condition, however it is unclear under what conditions they hold for the considered RNN class, not to mention deep SSMs considered in the paper at hand. The proof techniques both in Chen et al. (2020) and Mitarchuk et al. (2024) are different to what we employ.

**PAC bounds for Neural Ordinary Differential Equations.** PAC bounds for NODEs have been developed in Hanson et al. (2021a); Hanson & Raginsky (2024); Marion (2023); Fermanian et al. (2021). These results are based on Rademacher complexity and they are either affine in inputs or defined in the rough path sense. While a single block SSM interpreted in continuous time is affine in the input, general multi-block SSMs do not fall into this category. Moreover, these bounds are still exponential in the length of the integration interval, i.e., the length of the time series if fixed sampling time is used.

**PAC bounds for deep networks and transformers.** Trauger & Tewari (2024) derive a sequence-length-independent Rademacher complexity bound for single-layer transformers, improving slightly on Edelman et al. (2022) for multi-layer cases, though their bound grows logarithmically with the sequence length. However, their results do not apply to SSMs and involve matrix norms that may scale with the attention matrix size. Maintaining norm stability for longer sequences requires reducing certain matrix entries. In contrast, the $H_2/\ell_1$ norms in this paper depend only on state-space matrices and remain invariant to the input length.
Generalization bounds for deep neural networks (DNNs) extend beyond RNNs and dynamical systems (Bartlett et al., 2017; Liang et al., 2019; Golowich et al., 2018; Truong, 2022b). Since deep SSMs can simulate DNNs and resemble feedforward networks with fixed input length, their bounds should align with DNN results. Golowich et al. (2018) provide a depth-independent bound under bounded Schatten p-norm and a polynomial-depth bound for ReLU networks via contraction. Other works use spectral (Bartlett et al., 2017) or Fisher-Rao norms (Liang et al., 2019) to mitigate depth dependence. Truong (2022b) further refine Golowich et al. (2018) to a depth-independent, non-vacuous bound for non-ReLU activations. When applied to deep SSMs with trivial state-space components, the bounds of the present paper are more conservative than those of Golowich et al. (2018) for general activation functions, but are consistent with Golowich et al. (2018); Truong (2022b) for ReLU activation functions.

**PAC-Bayesian bounds for dynamical systems.** PAC-Bayesian bounds for various classes of dynamical systems were developed in Alquier & Wintenberger (2012); Alquier et al. (2013); Shalaeva et al. (2020); Haddouche & Guedj (2022a); Haussmann et al. (2021); Haddouche & Guedj (2022b); Seldin et al. (2012); Abeles et al. (2024)). The main difference between the cited papers and the present one are as follows.

1. *Single time-series vs. multiple independently sampled time-series.* All the cited papers assume that the data used for computing the empirical error is sampled from one single time series. The latter assumption required the use of various extensions of well-known concentration inequalities to the non-i.i.d. case. In particular, the obtained bounds all depend on some mixing coefficients. In contrast to the cited papers, the present paper assumes multiple i.i.d. samples of time-series', so formally, the learning problem of the present paper is completely different from the one of the papers cited above.

2. *PAC-Bayesian vs. PAC bounds.* The present paper presents a PAC bound, not a PAC-Bayesian one. PAC bounds have the advantage that they tend to be simpler to use and interpret, and they provide a uniform bound on the generalization gap, but they also tend to be fairly conservative. PAC-Bayesian bounds are more involved, they are sensitive to priors and they only bound the average (w.r.t. some posterior) generalization gap. However, they are potentially less conservative. This means that PAC bounds might actually be competitive with PAC-Bayesian ones in situations where the former is easy to evaluate and there are no obvious candidates for suitable priors. We believe that SSMs might fall in this category: the proposed PAC bound is easy to evaluate, and the choice of a suitable prior is far from obvious.

3. *Different model classes.* The classes of dynamical systems in Alquier & Wintenberger (2012); Alquier et al. (2013); Shalaeva et al. (2020); Haddouche & Guedj (2022a); Haussmann et al. (2021) do not

include state-space processes with partially observed state. The bound in Eringis et al. (2024) could, in principle, be applied to a one block SSM without non-linearities, and Eringis et al. (2023) can be applicable to multi-block SSMs in case the latter satisfies some stability conditions which are more stringent than the one in this paper. However, the application of Eringis et al. (2023; 2024) is possible only if the data used for learning is sampled from a single time series. The same is true for Abeles et al. (2024).

**Finite-sample bounds for dynamical systems.** In recent years there has been a significant interest in deriving bounds on the true loss for dynamical systems for particular learning algorithms (Oymak et al., 2019; Oymak & Ozay, 2022; Simchowitz et al., 2019; Lale et al., 2020; Foster & Simchowitz, 2020; Ziemann & Tu, 2022; Ziemann et al., 2022; Tsiamis & Pappas, 2019; Ziemann et al., 2024). However, most of these papers consider learning from one single time-series. Notable exceptions are Tu et al. (2024); Zheng & Li (2020); Sun et al. (2020), where bounds for the true risk for linear State-Space Models were derived. However, there the derived bound does not relate the empirical loss to the true one, and it is applicable only for linear dynamical systems, i.e., one block SSM. Moreover, the derived bound is specific to the learning algorithm employed. The latter is based on least-squares solution to linear regression, and it does not seem to be directly applicable to deep SSMs with non-linear blocks. In contrast, the results of the present paper are applicable to deep SSMs and to any learning algorithms.

**PAC bounds for non i.i.d. data.** There is a significant body of literature on PAC bounds involving Rademacher complexity (McDonald & Shalizi, 2017; Mohri & Rostamizadeh, 2008; Kuznetsov & Mohri, 2017) or other complexity measures for non i.i.d data, including data chosen from a single time-series. As it was mentioned above, in this paper we consider a different learning problem, namely, learning from multiple independently sampled time series, as opposed to one single time series. Moreover, the cited papers propose PAC bounds which involve various measures of the complexity of the parameterization, e.g., Rademacher complexity, but they do not dwell on estimating those measures for various classes of dynamical systems, such as SSMs.

We argue that existing methods and techniques used in proofs for simple RNNs or single layer SSMs are not sufficient to handle deep SSM architectures, see Remark 5.6 and 5.7 for the details.

## 3   Informal statement of the result

**Notation.** Unless stated otherwise, we denote scalars with lowercase characters, vectors with lowercase bold characters and matrices with uppercase characters. The symbol $\odot$ denotes the elementwise product of vectors, i.e., $x \odot y = (x_1 y_1, \ldots, x_n y_n)^T$ for all $x = (x_1, \ldots, x_n)^T, y = (y_1, \ldots, y_n)^T \in \mathbb{R}^n$. We use $[n]$ to denote the set $\{1, 2, \ldots, n\}$ for $n \in \mathbb{N}$. For a vector $v$ we denote by $v^{(j)}$ its $j$th coordinate. For vector valued time function $\mathbf{u}$, the notation $\mathbf{u}^{(j)}[t]$ refers to the $j$th coordinate of the value of function at time $t$. Furthermore, we use $\Sigma$ to denote a dynamical system specified in the context. The constant $n_{\text{in}}$ refers to the dimension of the input sequence, $T$ refers to its length in time, while $n_{\text{out}}$ is the dimension of the output (not necessarily a sequence). Denote by $\ell_T^2(\mathbb{R}^n)$ and $\ell_T^\infty(\mathbb{R}^n)$ the finite-dimensional Banach spaces generated by the all finite sequences over $\mathbb{R}^n$ of length $T$, viewed as vectors of $\mathbb{R}^{nT}$, with the Eucledian norm $\|\cdot\|_2$ and the supremum norm $\|\cdot\|_\infty$ over $\mathbb{R}^{nT}$ respectively. If $\mathcal{X}$ is a Banach space, we denote its norm by $\|\cdot\|_{\mathcal{X}}$. In particular, $\|\mathbf{u}\|_{\ell_T^2(\mathbb{R}^n)}^2 = \sum_{k=1}^T \|\mathbf{u}[k]\|_2^2$, and $\|\mathbf{u}\|_{\ell_T^\infty(\mathbb{R}^n)} = \sup_{k \in [T], j \in [n]} |\mathbf{u}^j[k]|$. For a Banach space $\mathcal{X}$, $B_{\mathcal{X}}(r) = \{x \in \mathcal{X} \mid \|x\|_{\mathcal{X}} \leq r\}$ denotes the ball of radius $r > 0$ centered at zero.

**Learning problem.** We consider the usual supervised learning framework for sequential input data. That is, we consider a family $\mathcal{F}$ of models, each model $f \in \mathcal{F}$ is a function which maps sequences of elements $\mathbf{u}[1], \ldots, \mathbf{u}[T]$ of the *input space* $\mathbb{R}^{n_{\text{in}}}$ to outputs (labels) in $\mathcal{Y} \subseteq \mathbb{R}^{n_{\text{out}}}$. We fix the length of the sequences to $T$ and we denote by $\mathcal{U}$ the set of all sequences of elements of $\mathbb{R}^{n_{\text{in}}}$ of length $T$. That is, $f \in \mathcal{F}$ can be viewed as a function $f : \mathcal{U} \to \mathcal{Y}$.

A dataset is an i.i.d sample of the form $S = \{(\mathbf{u}_i, \mathbf{y}_i)\}_{i=1}^N$ from some probability distribution $\mathcal{D}$, where $\mathbf{u}_i \in \mathcal{U}$ is a sequence of length $T$ having elements in $\mathbb{R}^{n_{\text{in}}}$, and $\mathbf{y}_i$ belongs to $\mathcal{Y}$. The probability measure determined by $\mathcal{D}$ is defined on the $\sigma$-algebra generated by the Borel sets of $\mathcal{U} \times \mathcal{Y} = \mathbb{R}^{n_{\text{in}}T} \times \mathcal{Y}$, where the set $\mathcal{U}$ of

sequences of elements of $\mathbb{R}^{n_{\text{in}}}$ of length $T$ is identified with $\mathbb{R}^{n_{\text{in}}T}$. We use the symbols $\mathbb{E}_{(\mathbf{u},\mathbf{y})\sim\mathcal{D}}$, $\mathbb{P}_{(\mathbf{u},\mathbf{y})\sim\mathcal{D}}$, $\mathbb{E}_{S\sim\mathcal{D}^N}$ and $\mathbb{P}_{S\sim\mathcal{D}^N}$ to denote expectations and probabilities w.r.t. a probability measure $\mathcal{D}$ and its $N$-fold product $\mathcal{D}^N$ respectively. The notation $S\sim\mathcal{D}^N$ tacitly assumes that $S\in(\mathcal{U}\times\mathcal{Y})^N$, i.e. $S$ is made of $N$ tuples of input sequences and output labels.

An elementwise loss function is a function $\ell:\mathcal{Y}\times\mathcal{Y}\to\mathbb{R}^+$ such that $\ell(\mathbf{y},\mathbf{y}')=0$ iff $\mathbf{y}=\mathbf{y}'$. Its role is to measure the discrepancy between predicted and true outputs (labels).

We define the *empirical loss* as $\mathcal{L}_{emp}^S(f) = \frac{1}{N}\sum_{i=1}^N \ell(f(\mathbf{u}_i),\mathbf{y}_i)$ and the *true loss* as $\mathcal{L}(f) = \mathbf{E}_{(\mathbf{u},\mathbf{y})\sim\mathcal{D}}[\ell(f(\mathbf{u}),\mathbf{y})]$. The goal of this paper is to bound the *generalization error or gap*, $\mathcal{L}(f)-\mathcal{L}_{emp}^S(f)$ uniformly for all models $f\in\mathcal{F}$.

We will be interested in model classes $\mathcal{F}$, elements of which arise by combining neural networks and the so-called State-Space Models.

**Model class of SSMs.** A *State-Space Model (SSM)* is a discrete-time, Linear Time-Invariant (LTI) dynamical system of the form

$$\Sigma\begin{cases} \mathbf{x}[k+1] = A\mathbf{x}[k] + B\mathbf{u}[k],\ \mathbf{x}[1]=0 \\ \mathbf{y}[k] = C\mathbf{x}[k] + D\mathbf{u}[k] \end{cases} \tag{1}$$

where $A\in\mathbb{R}^{n_x\times n_x}, B\in\mathbb{R}^{n_x\times n_u}, C\in\mathbb{R}^{n_y\times n_x}$ and $D\in\mathbb{R}^{n_y\times n_u}$ are matrices, $\mathbf{u}[k],\mathbf{x}[k]$ and $\mathbf{y}[k]$ are the input, the state and the output signals respectively for $k=1,2,\ldots,T$, where $T$ is the number of time steps. Here $n_x$ is the state dimension, $n_u$ is the input dimesion and $n_y$ is the output dimension of the SSM (LTI system). Note that $n_u$ and $n_y$ may not coincide with $n_{\text{in}}$ and $n_{\text{out}}$ as the former ones refer to the SSM's input and output, while the latter ones refer to the actual dataset. In this paper we are interested in *internally stable* (*stable* for short) SSMs, i.e. in SSMs for which the matrix $A$ from equation 1 is Schur. By definition, a matrix $A$ is Schur if its eigenvalues are inside the complex unit disk. This is a widespread definition in control theoretic literature (Antoulas, 2005). We remark that within other domains, a matrix being Schur may be defined in a different way which need not be equivalent to our definition.

Intuitively, stable SSMs are robust to perturbations, i.e., their state and output are continuous in the initial state and input, see for instance Antoulas (2005) for more details.

*Remark* 3.1 (Relationship between discrete-time SSMs (equation 1) and continuous-time SSMs). In the literature, the SSM layer is often defined as a continuous-time system

$$\dot{\mathbf{x}}_c(t) = A_c\mathbf{x}_c(t) + B_c\mathbf{v}(t),\ \mathbf{y}_c(t) = C\mathbf{x}_c(t) + D\mathbf{v}(t),\ \mathbf{x}_c(0)=0 \tag{2}$$

where $t\in[0,\infty)$. In order to transform equation 2 to a model mapping sequences to sequences, it is discretized in time (Gu et al., 2021a; 2022; Smith et al., 2022; Gu & Dao, 2023; Dao & Gu, 2024). That is, the following discrete-time system is considered:

$$\mathbf{x}[k+1] = A(\Delta_k)\mathbf{x}[k] + B(\Delta_k)\mathbf{u}[k],\ \mathbf{y}[k] = C\mathbf{x}[k] + D\mathbf{u}(k),\ \mathbf{x}[1]=0 \tag{3}$$

such that the matrix valued functions $A(\Delta)$ and $B(\Delta)$ are defined as $A(\Delta)=e^{A_c\Delta}$, $B(\Delta)=\int_0^\Delta e^{A(\Delta-s)}Bds$, $\Delta_k=\Delta(\mathbf{u}[k])$ is a function of $\mathbf{u}[k]$, and if $\mathbf{v}(t)=\mathbf{u}[k]$ for all $t\in(\sum_{i=1}^{k-1}\Delta_i,\sum_{i=1}^k\Delta_i)$, then $\mathbf{x}[k]=\mathbf{x}_c(\Delta_{k-1})$, $\mathbf{y}[k]=\mathbf{y}_c(\Delta_{k-1})$, $k\in[T]$, and $\Delta_0:=0$. If $\Delta_k$ equals a constant $\Delta$, then equation 3 describes an LTI system given by equation 1 with $A=A(\Delta)$ and $B=B(\Delta)$. Note that if $A_c$ is *Hurwitz*, i.e., all its eigenvalues have a negative real part, then $A(\Delta)$ is a Schur matrix for $\Delta>0$, i.e., the arising SSM block is stable. Also, $A_c$ being Hurwitz is equivalent to equation 2 being stable Antoulas (2005).

*Remark* 3.2 (Selective SSMs). If $\Delta$ in equation 3 depends on the input as in Gu & Dao (2023); Dao & Gu (2024), one obtains a discrete-time *Linear Parameter-Varying system (LPV)* (Tóth, 2010), or a so-called selective State-Space Model. In this case $A$ and $B$ (sometimes $C$ as well) depend on $\mathbf{u}[k]$ at each step. While they are more general, than LTI models and widely used in practice, they present greater analytical challenges. Extending our results to such models remains future work.

An SSM given by equation 1 induces a linear function $\mathcal{S}_{\Sigma,T}$ which maps every input sequence $\mathbf{u}[1],\ldots,\mathbf{u}[T]$ to the output sequence $\mathbf{y}[1],\ldots,\mathbf{y}[T]$. In particular, $\mathcal{S}_{\Sigma,T}$ has a well-defined induced norm as a linear operator,

defined in the usual way. For stable SSMs this norm can be bounded uniformly in $T$.

A *SSM block* is a residual composition of the SSM with a non-linear function $g$ applied element-wise, i.e. an SSM block maps the sequence $\mathbf{u}[1], \ldots, \mathbf{u}[T]$ to the sequence defined by $f^{\mathrm{DTB}}(\mathbf{u})[k] = g(\mathcal{S}_{\Sigma,T}(\mathbf{u})[k]) + \alpha \mathbf{u}[k]$, where $\alpha \in \mathbb{R}$ is the residual weight. A *deep SSM model* is a composition of several SSM blocks with an encoder, and a decoder transformation preceded by a time-pooling layer. That is, the input-output map of a deep SSM is a composition of functions of the form $f^{\mathrm{Dec}} \circ f^{\mathrm{Pool}} \circ f^{\mathrm{B}_L} \circ \ldots \circ f^{\mathrm{B}_1} \circ f^{\mathrm{Enc}}$, where $\circ$ denotes composition of functions. The functions $f^{\mathrm{Enc}}$ and $f^{\mathrm{Dec}}$ are linear transformations which are constant in time and are applied to sequences element-wise, while $f^{\mathrm{B}_i}$ is the input-output map of an SSM block for all $i$, $1 \le i \le L$, where $L$ is the depth of the model. This definition covers many examples from the literature, e.g. S4 (Gu et al., 2021a), S4D (Gu et al., 2022), S5 (Smith et al., 2022) or LRU (Orvieto et al., 2023).

The main result of this paper is the following **PAC bound for deep SSMs**:

**Theorem 3.3** (Informal theorem)**.** *Let $\mathcal{F}$ be a set of deep SSM models with stable SSM blocks, which satisfy a number of mild regularity assumptions. There exist constants $K_l$ and $K_{\mathcal{F}}$ which depend only on the model class $\mathcal{F}$, such that for any time horizon $T > 0$, any confidence level $1 > \delta > 0$, with probability at least $1 - \delta$ over the data sample $S \sim \mathcal{D}^N$,*

$$\forall f \in \mathcal{F} : \mathcal{L}(f) - \mathcal{L}^S_{emp}(f) \le \frac{K_{\mathcal{F}} + 4K_l \sqrt{2\log\left(\frac{4}{\delta}\right)}}{\sqrt{N}} \tag{4}$$

With standard arguments on PAC bounds and Rademacher complexity, the result above also implies the following oracle inequality for the Empirical Risk Minimization framework (Shalev-Shwartz & Ben-David, 2014).

**Corollary 3.4.** *With the assumptions of Theorem 3.3 for $f_{ERM} = \mathrm{argmin}_{f \in \mathcal{F}} \mathcal{L}^{\mathcal{S}}_{emp}(f)$, for any $1 > \delta > 0$, with probability at least $1 - \delta$ over the data sample $S \sim \mathcal{D}^N$,*

$$\mathcal{L}(f_{\mathrm{ERM}}) \le \min_{f \in \mathcal{F}} \mathcal{L}(f) + \frac{K_{\mathcal{F}} + 5K_l \sqrt{2\log\left(\frac{8}{\delta}\right)}}{\sqrt{N}} \tag{5}$$

Bounds given by inequalities 4 and 5 ensure that as $N$ grows, the empirical and true losses converge, and the learned model's true loss approaches the minimum possible loss.

The term $K_{\mathcal{F}}$ depends on the norms of the SSM blocks and the magnitudes of non-SSM weights, but it remains independent of $T$. Since in general, norms of SSMs decrease as their stability increases, *stability makes the generalization gap insensitive to sequence length, and increasing stability further decreases it.*

Specifically, for deep SSMs with $k$ layers, $K_{\mathcal{F}} = O((\text{SSM norm})^k (\text{non-SSM weight norm})^k)$. While $K_{\mathcal{F}}$ grows exponentially with the depth unless all components are contractions, high non-SSM weights can be offset by lower SSM norms. These norms decrease as SSMs become more stable, though stability is not directly tied to weight magnitudes — stable SSMs can still have large weights. This exponential dependence aligns with bounds for deep neural network (Golowich et al., 2020; Truong, 2022a).

Depth may negatively impact the generalization gap, but this does not imply poor generalization overall. Even if $K_{\mathcal{F}}$ is large for deep SSMs, inequality 5 implies that if the best true error is small then the generalization gap can still be small. Additionally, as $N$ increases, the influence of $K_{\mathcal{F}}$ diminishes, suggesting deeper models require more data, which is consistent with findings on deep neural networks.

## 4 Formal problem setup

### 4.1 Rademacher complexity

Our main result is essentially an upper bound on the Rademacher complexity of a set of deep SSMs with specific properties, thus we begin by recalling the definition.

**Definition 4.1** (Def. 26.1 in Shalev-Shwartz & Ben-David (2014))**.** The Rademacher complexity of a bounded set $\mathcal{A} \subset \mathbb{R}^m$ is defined as

$$R(\mathcal{A}) = \mathbb{E}_{\boldsymbol{\sigma}} \left[ \sup_{a \in \mathcal{A}} \frac{1}{m} \sum_{i=1}^{m} \sigma_i a_i \right],$$

where the random variables $\sigma_i$ are i.i.d such that $\mathbb{P}(\sigma_i = 1) = \mathbb{P}(\sigma_i = -1) = 0.5$ and $\boldsymbol{\sigma} = (\sigma_1, \dots, \sigma_m)^T$. The Rademacher complexity of a set of functions $\mathcal{H}$ defined over $\mathcal{U} \times \mathcal{Y}$, with respect to the sample $S = ((\mathbf{u}_1, \mathbf{y}_1) \dots, (\mathbf{u}_m, \mathbf{y}_m)) \in (\mathcal{U} \times \mathcal{Y})^N$ is defined as

$$R_S(\mathcal{H}) = R(\{(h(\mathbf{u}_1, \mathbf{y}_1), \dots, h(\mathbf{u}_m, \mathbf{y}_m))^T \mid h \in \mathcal{H}\}) = \mathbb{E}_{\sigma} \left[ \sup_{h \in \mathcal{H}} \frac{1}{m} \sum_{i=1}^{m} \sigma_i h(\mathbf{u}_i, \mathbf{y}_i) \right].$$

Intuitively, the Rademacher complexity $R_S(\mathcal{H})$ tries to capture the sensitivity of $\mathcal{H}$ to overfit random noise. The term $\sum_{i=1}^{m} \sigma_i h(\mathbf{u}_i, \mathbf{y}_i)$ can be seen as an inner product, therefore the supremum over $\mathcal{H}$ has the intuitive meaning of finding the function $h \in \mathcal{H}$ that aligns best with the random vector $\boldsymbol{\sigma}$. In this context, the alignment is measured with an inner product as opposed to the usual cosine similarity, where the inner product is inversely scaled with the norm of the vectors. Practically, the ability to align with a random vector correlates with the ability to overfit the data. The expectation over $\boldsymbol{\sigma}$ means we consider the average (over the random vectors) maximum alignment between the function outputs on the data sample and the random vector.

A different intuitive explanation of the above definition can be found in Section 26.1 of Shalev-Shwartz & Ben-David (2014). In a nutshell, this second explanation is that the sum $\frac{1}{m} \sum_{i=1}^{m} \sigma_i h(\mathbf{u}_i, \mathbf{y}_i) = \frac{1}{m} \sum_{i=1,\dots,m,\sigma_i=1} h(\mathbf{u}_i, \mathbf{y}_i) - \frac{1}{m} \sum_{i=1,\dots,m,\sigma_i=-1} h(\mathbf{u}_i, \mathbf{y}_i)$ represents the difference between the average performance of the model on the validation dataset (data points for which $\sigma_i = 1$) and on the training dataset (data points for which $\sigma_i = -1$), when the elements of both datasets are chosen randomly. Then the expectation over sigma of the supremum of these sums is intuitively proportional to the worst-case generalization gaps (difference between the true loss and the empirical loss).

The following is a standard generalization theorem, involving Rademacher complexity, that we build our proof on.

**Theorem 4.2** (Theorem 26.5 in Shalev-Shwartz & Ben-David (2014))**.** *Let $L_0$ denote the set of functions of the form $(\mathbf{u}, \mathbf{y}) \mapsto \ell(f(\mathbf{u}), \mathbf{y})$ for $f \in \mathcal{F}$. Let $K_l$ be such that the functions from $L_0$ all take values from the interval $[0, K_l]$. Then for any $\delta \in (0, 1)$ we have*

$$\mathbb{P}_{S \sim \mathcal{D}^N} \left( \forall f \in \mathcal{F} : \mathcal{L}(f) - \mathcal{L}_{emp}^S(f) \leq 2R_S(L_0) + 4K_l \sqrt{\frac{2 \log\left(\frac{4}{\delta}\right)}{N}} \right) \geq 1 - \delta.$$

### 4.2 Deep SSMs

**Stable SSMs.** In the sequel, we consider solutions of equation 1 on the time interval $[1, T]$, where the value of $T$ is fixed. As it was mentioned in Section 3, we consider only LTI systems given by equation 1, for which the matrix $A$ is Schur. It is well-known (Antoulas, 2005) that (internal) stability is equivalent to the $A$ matrix in equation 1 being Schur, i.e., meaning all the eigenvalues of $A$ are inside the complex unit disk. In particular, a sufficient, but not necessary condition for stability is that $A$ is a contraction, i.e. $\|A\|_2 < 1$. Moreover, for a system given by equation 1 with $A$ being a Schur matrix, there exists a non-singular matrix $P$ representing a linear basis transformation such that the transformed system given by $A' = PAP^{-1}, B' = PB, C' = CP^{-1}, D' = D$ has the same Markov parameters as the original one (namely $CA^k B = C'(A')^k B'$ for all $k \in \mathbb{N}$) and $\|A'\|_2 < 1$. Markov parameters of an LTI system are the members of the set $\{D\} \cup \{CA^k B \mid k \in \mathbb{N}\}$ and are widely used in control theory. In fact, the output of an LTI system can be formulated as a function of the input sequence and its Markov parameters, namely $\mathbf{y}[k] = D\mathbf{u}[k] + \sum_{i=1}^{k} CA^{i-1} B\mathbf{u}[k-i]$ for $0 \leq k \leq T$.

All popular architectures in the literature use stable SSM blocks, see Table 4.2.

If equation 2 describes a stable continuous-time linear system, i.e. $A_c$ is a Hurwitz matrix (all the eigenvalues of $A_c$ have a negative real part), then $A(\Delta)$ is a Schur matrix (Antoulas, 2005), i.e., the corresponding discrete-time SSM is stable.

**Input-output maps of SSMs as operators on $\ell_T^p$, $p = \infty, 2$.** As it was mentioned in Section 3, an SSM given by equation 1 induces an input-output map $\mathcal{S}_{\Sigma,T}$, which maps every input sequence $\mathbf{u}[1], \ldots, \mathbf{u}[T]$ to output sequence $\mathbf{y}[1], \ldots, \mathbf{y}[T]$, and can be described by a convolution $\mathbf{y}[t] = \mathcal{S}_{\Sigma}(\mathbf{u})[t] = \sum_{j=1}^{t} H_{j-1}\mathbf{u}[t-j+1]$, where $H_0 = D$ and $H_j = CA^{j-1}B$, $j > 0$. The map $\mathcal{S}_{\Sigma,T}$ can be viewed as a linear operator $\mathcal{S}_{\Sigma,T} : \ell_T^p(\mathbb{R}^{n_u}) \to \ell_T^\infty(\mathbb{R}^{n_y})$, for any choice $p \in \{\infty, 2\}$. In particular, $\mathcal{S}_{\Sigma,T}$ has a well-defined induced norm as a linear operator, defined in the usual way,

$$\|\mathcal{S}_{\Sigma,T}\|_{\infty,p} = \sup_{\mathbf{u} \in \ell_T^p(\mathbb{R}^{n_u})} \frac{\|\mathcal{S}_{\Sigma,T}(\mathbf{u})\|_{\ell_T^\infty(\mathbb{R}^{n_y})}}{\|\mathbf{u}\|_{\ell_T^p(\mathbb{R}^{n_y})}}.$$

| Model | SSM | Block |
|---|---|---|
| S4 (Gu et al., 2021a) | LTI, $A_c = \Lambda - PQ^*$ block-diagonal, **stable** | SSM + nonlinear activation |
| S4D (Gu et al., 2022) | LTI, $A_c = -\exp(A_{Re}) + i \cdot A_{Im}$ block-diagonal, **stable** | SSM + nonlinear activation |
| S5 (Smith et al., 2022) | LTI, **stable** diagonal $A_c$ | SSM + nonlinear activation |
| LRU (Orvieto et al., 2023) | LTI, diagonal $A_c$ **stable** complex exponential parametrization | SSM + MLP / GLU + skip connection |

Table 1: Summary of popular deep SSM models. The SSM blocks in these models arise by discretizing a continuous-time system, see Remark 3.1.

It is a standard result in control theory that if $\Sigma$ is internally stable, the supremum $\|\Sigma\|_{\infty,p} = \sup_{T>0} \|\mathcal{S}_{\Sigma,T}\|_{\infty,p}$ of these norms is finite, see Antoulas (2005). In this paper, we will use upper bounds on the induced norms $\|\mathcal{S}_{\Sigma,T}\|_{r,\infty}$, $r \in \{2, \infty\}$ to bound the Rademacher complexity. In turn, these norms can be upper bounded by the following two standard control-theoretical norms defined on SSMs. For an SSM $\Sigma$ given by equation 1 let us define the $\ell_1$ (Chellaboina et al., 1999) and $H_2$ (Antoulas, 2005) norms of $\Sigma$, denoted by $\|\Sigma\|_1$ and $\|\Sigma\|_2$ respectively, as

$$\|\Sigma\|_1 := \max_{1 \le i \le n_y} \left[ \|D_i\|_1 + \sum_{k=0}^{\infty} \|C_i A^k B\|_1 \right],$$

$$\|\Sigma\|_2 := \sqrt{\|D\|_F^2 + \sum_{k=0}^{\infty} \|CA^k B\|_F^2}.$$

**Lemma 4.3** (Chellaboina et al. (1999)). *For a system given by equation 1, it holds that* $\sup_{T\ge0} \|\mathcal{S}_{\Sigma,T}\|_{2,\infty} \le \|\Sigma\|_1$ *and* $\sup_{T\ge0} \|\mathcal{S}_{\Sigma,T}\|_{\infty,\infty} \le \|\Sigma\|_2$.

An upper bound on the norms $\|\Sigma\|_i$, $i = 1, 2$ can be easily computed by solving a suitable a linear matrix inequality (LMI), which is a standard tool in control theory (Boyd et al., 1994). Moreover, $\|\Sigma\|_2$ can also be computed using Sylvester equations, for which standard numerical algorithms exist (Antoulas, 2005). Alternatively, both norms can be computed by taking a sufficiently large finite sum instead of

the infinite sum used in their definition. Finally, if $\|A\|_2 < \beta < 1$, then an easy calculation reveals that $\|\Sigma\|_1 \leq \left( \|D\|_2 + \frac{\|B\|_2 \|C\|_2}{1-\beta} \right)$ and $\|\Sigma\|_2 \leq \sqrt{\|D\|_F^2 + \frac{n_y \|B\|_2^2 \|C\|_2^2}{1-\beta^2}}$.

**Deep SSM models.** In this paper, we consider deep SSM models, which consist of layers of blocks, each block representing an SSM followed by a nonlinear transformation (MLP, Gated Linear Units (GLU), defined in Definiton 4.5). Moreover, these blocks are preceded by a linear encoder and succeeded by a pooling block and a linear decoder.

In order to define deep SSMs, first we define MLP and GLU layers. Then we define SSM blocks, which are compositions of SSMs given by equation 1 with MLP and GLU layers. Finally, we define deep SSM models, where all these elements are combined.

**Definition 4.4** (MLP layer). An MLP layer is a function $f : \ell_T^\infty(\mathbb{R}^{n_y}) \to \ell_T^\infty(\mathbb{R}^{n_u})$ such that there exist an integer $M \geq 1$, matrices and vectors $\{W_i, \mathbf{b}_i\}_{i=1}^{M+1}$ and activation function $\rho : \mathbb{R} \to \mathbb{R}$, such that $W_i \in \mathbb{R}^{n_{i+1} \times n_i}$ and $\mathbf{b} \in \mathbb{R}^{n_i}$, $i \in [M]$, $n_1 = n_y$ and $n_{M+1} = n_u$, and

$$f(\mathbf{u})[k] = g_{W_{M+1}, \mathbf{b}_{M+1}} \circ f_{W_M, \mathbf{b}_M} \circ \ldots \circ f_{W_1, \mathbf{b}_1}(\mathbf{u}[k]) \tag{6}$$

where $k \in [T]$, $f_{W_i, \mathbf{b}_i}(x) = \rho(g_{W_i, \mathbf{b}_i}(\mathbf{x}))$ and $g_{W_i, \mathbf{b}_i}(\mathbf{x}) = W_i \mathbf{x} + \mathbf{b}_i$ for all $i \in [M+1]$. By slightly abusing the notation, for a vector $\mathbf{v}$, $\rho(\mathbf{v})$ denotes the elementwise application of $\rho$ to $\mathbf{v}$.

Intuitively, a MLP layer represents a deep neural network applied to a signal at every time step. The function $f_{W_i, \mathbf{b}_i}$ represents the $i$th layer of this neural network, with activation function $\rho$ and weights $W_i, \mathbf{b}_i$. For the sake of simplicity, activation function is assumed to be the same across all layers of the neural network.

**Definition 4.5** (GLU layer (Smith et al., 2022)). A GLU layer is a function of the form $f : \ell_T^\infty(\mathbb{R}^{n_y}) \to \ell_T^\infty(\mathbb{R}^{n_u})$ parametrized by a matrix $W$ such that

$$f(\mathbf{u})[k] = GELU(\mathbf{u}[k]) \odot \sigma(W \cdot GELU(\mathbf{u}[k])), \tag{7}$$

where $\sigma$ is the sigmoid function, i.e. $\sigma(x) = (1 + e^{-x})^{-1}$, and GELU is the Gaussian Error Linear Unit (Hendrycks & Gimpel, 2016), namely $GELU(x) = x\Phi(x)$, where $\Phi(x) = \frac{1}{\sqrt{2\pi}} \int_{-\infty}^x e^{-s^2/2} \, ds$ is the cumulative distribution function of the Gaussian standard normal distribution. Analogously to Definition 4.4, for a vector $\mathbf{v}$, $\sigma(\mathbf{v})$ and $GELU(\mathbf{v})$ denote the elementwise application of $\sigma$ and $GELU$ to $\mathbf{v}$, respectively.

Note that this definition of GLU layer differs from the original definition in Dauphin et al. (2017), because in deep SSM models GLU is usually applied individually for each time step, without any time-mixing operations. See Appendix G.1 in Smith et al. (2022).

Next, we define a SSM block, which is a composition of an SSM layer with a MLP/GLU layer.

**Definition 4.6.** An SSM block is a function $f^{\mathrm{DTB}} : \ell_T^r(\mathbb{R}^{n_u}) \to \ell_T^\infty(\mathbb{R}^{n_u})$, $r \in \{2, \infty\}$, such that for all $k \in [T]$

$$f^{\mathrm{DTB}}(\mathbf{u})[k] = g \circ \mathcal{S}_{\Sigma, T}(\mathbf{u})[k] + \alpha \mathbf{u}[k] \tag{8}$$

for some SSM $\Sigma$ given by equation 1, some MLP or GLU layer $g : \ell_T^\infty(\mathbb{R}^{n_y}) \to \ell_T^\infty(\mathbb{R}^{n_u})$ and constant $\alpha$.

We incorporate $\alpha$ so that the definition covers residual connections (typically $\alpha$ is either 1 or 0). The definition above is inspired by the series of popular architectures mentioned in the introduction.

Finally, following the literature on SSMs, we define a deep SSM model as a composition of SSM blocks along with linear layers (encoder/decoder) combined with a time-pooling layer in case of classification.

**Definition 4.7** (encoder, decoder, pooling). An encoder is a function $f : \ell_T^p(\mathbb{R}^{n_{\mathrm{in}}}) \to \ell_T^p(\mathbb{R}^{n_u})$, where $p \in \{2, \infty\}$ is an integer, such that there exists a matrix $W_{\mathrm{Enc}}$ for which $f(\mathbf{u})[k] = W_{\mathrm{Enc}} \mathbf{u}[k]$. A decoder is a function $f : \mathbb{R}^{n_u} \to \mathbb{R}^{n_{\mathrm{out}}}$ such that there exists a matrix $W_{\mathrm{Dec}}$ such that $f(\mathbf{x}) = W_{\mathrm{Dec}} \mathbf{x}$. A pooling layer is the function $f^{\mathrm{Pool}} : \ell_T^\infty(\mathbb{R}^{n_u}) \to \mathbb{R}^{n_u}$ defined by $f^{\mathrm{Pool}}(\mathbf{u}) = \frac{1}{T} \sum_{k=1}^T \mathbf{u}[k]$.

An encoder corresponds to applying linear transformations to each element of the input sequence. The pooling layer is typically an average pooling over the time axis.

**Definition 4.8.** A deep SSM model is a function $f : \ell_T^2(\mathbb{R}^{n_{\text{in}}}) \to \mathbb{R}^{n_{\text{out}}}$ of the form

$$f = f^{\text{Dec}} \circ f^{\text{Pool}} \circ f^{\text{B}_L} \circ \ldots \circ f^{\text{B}_1} \circ f^{\text{Enc}} \tag{9}$$

where $f^{\text{Enc}}$ is an encoder and $f^{\text{Dec}}$ is a decoder, and $f^{\text{B}_i}$ are SSM blocks for all $i$, and $f^{\text{Pool}}$ is the pooling layer.

Definition 4.8 covers many important architectures from the literature, e.g. S4, S4D, S5 and LRU. Note that we did not include such commonly used normalization techniques as batch normalization in the definition since they are not relevant for our results. Indeed, once the model training is finished, a batch normalization layer corresponds to applying a neural network with linear activation function, i.e., it can be integrated into one of the neural network layers. Since the objective of PAC bounds is to bound the generalization error for already trained models, for the purposes of PAC bounds, normalization layers can be viewed as an additional layer of neural network.

*Remark* 4.9. The usual definition of a single layer, *simple* RNN (e.g. Chen et al. (2020)) is

$$\begin{cases} \mathbf{h}_{k+1} &= \sigma_1(U\mathbf{x}_k + W\mathbf{h}_k + \mathbf{b}) \\ \mathbf{y}_k &= \sigma_2(V\mathbf{h}_k + \mathbf{c}) \end{cases} \tag{10}$$

where $(U, W, V, \mathbf{b}, \mathbf{c})$ are the parameters of the RNN and $\sigma_1, \sigma_2$ are some fixed activation functions.

A single SSM block (Definition 4.6) is made of a single, linear SSM layer followed by a time independent nonlinearity. The linear SSM layer can be represented by a simple RNN. However, even a special case of such a SSM block would result in an RNN for which the activation functions $\sigma_1$ and $\sigma_2$ in equation 10 are different. Using MLPs or deep stack of such SSM blocks would result in a dynamical system whose structure is completely different from RNNs. Note that the MLP cannot be viewed as an activation function in equation 10: unlike the fixed activation function in equation 10, the MLP in the deep SSM is not part of the time-mixing component and it is parametrized, i.e. it is learned. Furthermore, as SSM layers are discrete-time LTI systems, they are invariant under linear state-space transformations, whereas simple RNNs are not.

## 4.3 Assumptions

Before moving forward to discuss the main result, we summarize the assumptions we make in the paper for the sake of readability.

**Assumption 4.10.** We consider a family $\mathcal{F}$ of deep SSM models of depth $L$ such that the following hold:

1. **Architecture.**

   There exist families $\mathcal{F}_{\text{Enc}}$ of encoders, $\mathcal{F}_{\text{Dec}}$ of decoders, $\mathcal{E}$ of SSMs, $\mathcal{F}_i$, $i \in [L]$, of nonlinear blocks, and collection of residual weights $\{\alpha_i\}_{i=1}^L$ such that if $f \in \mathcal{F}$ given by equation 9, then
   **(1)** the encoder $f^{\text{Enc}}$ belongs to $\mathcal{F}_{\text{Enc}}$, the decoder $f^{\text{Dec}}$ belongs to $\mathcal{F}_{\text{Dec}}$,
   **(2)** and if the $i$th SSM block $f^{\text{B}_i}$ is given by equation 8, then $\Sigma \in \mathcal{E}$, $\alpha = \alpha_i$, and $g$ belongs to $\mathcal{F}_i$.

2. **Scalar output.**

   Let $n_{\text{out}} = 1$.

3. **Lipschitz loss function.**

   Let the elementwise loss $\ell$ be $L_\ell$-Lipschitz continuous, i.e., $\ell(y_1, y_1') - \ell(y_2, y_2') \leq L_\ell(|y_1 - y_2| + |y_1' - y_2'|)$ for all $y_1, y_2, y_1', y_2' \in \mathbb{R}$.

4. **Bounded input.**

   There exist $K_{\mathbf{u}} > 0$ and $K_y > 0$ such that for any input trajectory $\mathbf{u}$ and label $y$ sampled from $\mathcal{D}$, with probability 1 we have that $\|\mathbf{u}\|_{\ell_T^2(\mathbb{R}^{n_{\text{in}}})} \leq K_{\mathbf{u}}$ and $|y| \leq K_y$.

5. **Stability & bounded encoder and decoder norms.**

   **(1)** Each element $\Sigma$ of $\mathcal{E}$ is stable, i.e., if $\Sigma$ is of the form equation 1, then $A$ is a Schur matrix. Moreover, there exist constants $K_1$ and $K_2$ such that $\|\Sigma\|_p \leq K_p$, $p = 1, 2$ for each $\Sigma \in \mathcal{E}$.
   **(2)** There exists constants $K_{\text{Enc}}, K_{\text{Dec}}$ such that if $f \in \mathcal{F}_{\text{Enc}}$, and $f(\mathbf{u})[k] = W_{\text{Enc}}\mathbf{u}[k]$ for a matrix $W_{\text{Enc}}$, then $\|W_{\text{Enc}}\|_{2,2} < K_{\text{Enc}}$, and if $f \in \mathcal{F}_{\text{Dec}}$ and $f(\mathbf{x}) = W_{\text{Dec}}\mathbf{x}$, then $\|W_{\text{Dec}}\|_{2,2} < K_{\text{Dec}}$.

6. **Nonlinear blocks are either MLP or GLU.**

   For every $i \in [L]$, $\mathcal{F}_i$ is either a family of MLP layers or a family of GLU layers. In the former case, all elements of $\mathcal{F}_i$ are MLP layers with $M_i$ layers and with the same activation functions $\rho_i$ which is either ReLU or a sigmoid-like function which satisfies the following: $\rho_i(0) = 0.5$, it is 1-Lipschitz, $\rho(x) \in [-1, 1]$, $\rho_i(x) - \rho_i(0)$ is odd for any $x$ in the domain of $\rho$. If $\mathcal{F}_i$ is a family of GLU layers, then each $\sigma_i$ is the actual sigmoid function, i.e. $\sigma(x) = (1 + e^{-x})^{-1}$.

7. **Bounded weights for MLP.**

   For every $i \in [L]$ such that $\mathcal{F}_i$ is a family of MLP layers, there exists $K_{W,i}, K_{\mathbf{b},i}$ such that for every $f \in \mathcal{F}_i$ given by equation 6 with $M = M_i$ and $\rho = \rho_i$, the weights of $f$ satisfy

   $$\max_{j \in [M+1]} \|W_j\|_{\infty,\infty} < K_{W,i}, \quad \max_{j \in [M+1]} \|\mathbf{b}_j\|_{\infty} < K_{\mathbf{b},i}.$$

8. **Bounded weights for GLU.**

   For every $i \in [L]$ such that $\mathcal{F}_i$ is a family of GLUs, there exists a constant $K_{\text{GLU},i}$ such that for every $f \in \mathcal{F}_i$ given by equation 7 with $\sigma = \sigma_i$, $\|W_i\|_{\infty,\infty} < K_{\text{GLU,i}}$.

The first assumption is a standard one, the only restriction is that all deep SSMs have the same depth and all SSM blocks have the same residual connection.

Assumption 2, though being restrictive, covers key scenarios such as classification and 1-dimensional regression, which are central to theoretical analysis.

Assumption 3 requires the loss function to be Lipschitz-continuous, which is a standard assumption in machine learning and holds for most of the loss functions used in practice, including the squared loss on bounded domains, the $\ell_1$ loss and the cross-entropy loss (see Appendix A). This ensures boundedness during the learning process.

Assumption 4 is also fairly standard, as input normalization is common in practice.

Assumption 5 is the key assumption enforcing SSM stability via structured parametrization. Beyond numerical benefits, stability ensures reliable predictions by preventing small input changes from causing large output variations, crucial for learning and inference. Many prior work mentioned in Section 2 proving PAC bounds for simple RNNs assume that $\|U\|_2 < 1$ for $U$ in equation 10, corresponding to $\|A\|_2 < 1$ in equation 1. In contrast, we require the system norm to be finite that is achieved by assuming that the SSM layers are stable, i.e. the state matrix $A$ is Schur in every layer. As stated in Section 4.2, this does not necessarily imply that $\|A\|_2 < 1$. The converse holds, namely $\|A\|_2 < 1$ implies stability.

Assumptions 6 and 7 are again considered standard, requiring non-linear layers to be either all MLPs or all GLUs with specific activations and enforcing bounded weights for the encoder, decoder, and MLP/GLU layers.

## 5 Main results

We derive a Rademacher complexity-based generalization bound for deep SSM models, independent of sequence length. The key challenges are:
**(1)** bounding the Rademacher complexity of SSMs, **(2)** extending this to hybrid SSM-neural network blocks, and **(3)** handling deep architectures with multiple such blocks. For stable SSMs, we show their norm bounds the Rademacher complexity for any sequence length. To address the second and third challenges, we introduce *Rademacher Contraction*, a universal framework that enables componentwise complexity estimation in deep models.

**Definition 5.1** (($\mu, c$)-*Rademacher Contraction*). Let $X_1$ and $X_2$ be subsets of Banach spaces $\mathcal{X}_1, \mathcal{X}_2$, with norms $\| \cdot \|_{\mathcal{X}_1}$ and $\| \cdot \|_{\mathcal{X}_2}$, and let $\mu \geq 0$ and $c \geq 0$. A set of functions $\Phi = \{\varphi : X_1 \to X_2\}$ is said to be ($\mu, c$)-*Rademacher Contraction*, or ($\mu, c$)-RC in short, if for all $N \in \mathbb{N}^+$ and $Z \subseteq X_1^N$ we have

$$\mathbb{E}_{\boldsymbol{\sigma}} \left[ \sup_{\varphi \in \Phi} \sup_{\{\mathbf{u}_i\}_{i=1}^N \in Z} \left\| \frac{1}{N} \sum_{i=1}^N \sigma_i \varphi(\mathbf{u}_i) \right\|_{\mathcal{X}_2} \right] \leq \mu \mathbb{E}_{\boldsymbol{\sigma}} \left[ \sup_{\{\mathbf{u}_i\}_{i=1}^N \in Z} \left\| \frac{1}{N} \sum_{i=1}^N \sigma_i \mathbf{u}_i \right\|_{\mathcal{X}_1} \right] + \frac{c}{\sqrt{N}}, \tag{11}$$

where $\sigma_i$ are i.i.d. Rademacher random variables, $i \in [N]$, i.e. $\mathbb{P}(\sigma_i = 1) = \mathbb{P}(\sigma_i = -1) = 0.5$ and $\boldsymbol{\sigma} = (\sigma_1, \ldots, \sigma_N)^T$.

**Rademacher Contractions in the literature.** While the concept of RC is new, special cases of Definition 5.1 have been used in the literature for bounding Rademacher complexity of deep neural networks. In Golowich et al. (2018) the authors considered biasless ReLU networks and proved a similar inequality using Talagrand's Contraction Lemma (Ledoux & Talagrand, 1991). In Truong (2022b), the author considered neural networks with dense and convolutional layers and derived a PAC bound via bounding the Rademacher complexity. One of the key technical achievements in Truong (2022b) is Theorem 9, which is a more general version of the inequality in Golowich et al. (2018). This was then applied to obtain generalization bounds for the task of learning Markov-chains in Truong (2022a), however the generalization error was measured via the marginal cost and the ($\mu, c$)-RC type inequality was only applied for time-invariant neural networks. In contrast, we prove that along with time invariant models, stable SSMs, defined between certain Banach spaces, also satisfy inequality 11 and apply it to deep structures.

In a recent work Trauger & Tewari (2024), the authors consider Transformers and implicitly establish similar inequalities to inequality 11 by bounding different kinds of operator norms of the model and managed to extend it to a stack of Transformer layers. Besides these similarities, some key differences in our work are that Definition 5.1 provides an explicit way to combine SSMs with neural networks, even in residual blocks; we do not assume the SSM matrices to be bounded, instead we require the system norm to be bounded via stability, which is a weaker condition; and we upper bound the Rademacher complexity directly instead of bounding the covering number.

**Interpretation of the ($\mu, c$)-RC inequality.** Inequality 11 allows relating the Rademacher complexity of a model class to the Rademacher complexity of its inputs via the constants $\mu$ and $c$. These constants depend on the model class as well as the domain $X_1$ and range $X_2$ of the models. As shown next, the RC property is preserved under the composition of layers.

**Lemma 5.2** (Composition lemma). *Let* $\Phi_1 = \{\varphi_1 : X_1 \to X_2\}$ *be* ($\mu_1, c_1$)-*RC and* $\Phi_2 = \{\varphi_2 : X_2 \to X_3\}$ *be* ($\mu_2, c_2$)-*RC. Then the set of compositions* $\Phi_2 \circ \Phi_1 := \{\varphi_2 \circ \varphi_1 \mid \varphi_1 \in \Phi_1, \varphi_2 \in \Phi_2\}$ *is* ($\mu_1 \mu_2, \mu_2 c_1 + c_2$)-*RC.*

The proof is in Appendix B. Consequently, for deep models composed of layers that each satisfy the RC property, the entire model class is RC as well. Then inequality 11 can be applied to bound the Rademacher complexity of the deep model by that of the input sample. The latter can often be bounded, for instance:

**Lemma 5.3.** $\mathbb{E}_{\boldsymbol{\sigma}} \left[ \left\| \frac{1}{N} \sum_{i=1}^N \sigma_i \mathbf{u}_i \right\|_{\ell_T^2(\mathbb{R}^{n_{\text{in}}})} \right] \leq \frac{K_{\mathbf{u}}}{\sqrt{N}}$ *for all* $\|\mathbf{u}_i\| \in B_{\ell_T^2(\mathbb{R}^{n_{\text{in}}})}(K_{\mathbf{u}})$, $i \in [N]$.

The proof follows a standard argument, e.g. see Lemma 26.10 in Shalev-Shwartz & Ben-David (2014), for completeness it is presented in Appendix B.

That is, in order to bound the Rademacher complexity of deep SSMs, all we need to show is that each component of a deep SSM model is ($\mu, c$)-RC for some $\mu$ and $c$ with compatible domains and ranges. To this end, for each $i \in [L]$, define the *family* $\mathcal{F}_i^{\text{DTB}}$ *of ith SSM blocks* as the family of all SSMs blocks $f^{\text{DTB}}$ given by equation 8 such that $g \in \mathcal{F}_i$, $\Sigma \in \mathcal{E}$, $\alpha = \alpha_i$. In particular, for any $f \in \mathcal{F}$ given by equation 9, the $i$th SSM block $f^{\text{B}_i}$ belongs to $\mathcal{F}_i^{\text{DTB}}$.

**Lemma 5.4.** *For each set* $\mathcal{X}$ *of model layers interpreted as functions between Banach spaces such that* $\mathcal{X} \in \{\mathcal{F}_{\text{Enc}}, \mathcal{F}_{\text{Dec}}, \mathcal{F}_i, \mathcal{F}_i^{\text{DTB}}, i \in [L], \mathcal{E}, \{f^{\text{Pool}}\}\}$, *the set* $\mathcal{X}|_{B(r)}$, *i.e. the elements of* $\mathcal{X}$ *restricted to a ball of radius* $r$, *is* ($\mu_{\mathcal{X}}(r), c_{\mathcal{X}}(r)$)-*RC in their domain, and the range of the elements of* $\mathcal{X}|_{B(r)}$ *is a subset of*

| LAYER TYPE $\mathcal{X}$, $\mathcal{X} \neq \mathcal{F}_i$ | $\mu_{\mathcal{X}}(r)$ | $c_{\mathcal{X}}(r)$ | $\hat{r}_{\mathcal{X}}(r)$ |
|---|---|---|---|
| $\mathcal{X} = \mathcal{F}_{\mathrm{Enc}}$ | $K_{\mathrm{Enc}}$ | $0$ | $K_{\mathrm{Enc}}r$ |
| $\mathcal{X} = \mathcal{F}_{\mathrm{Dec}}$ | $K_{\mathrm{Dec}}$ | $0$ | $K_{\mathrm{Dec}}r$ |
| $\mathcal{X} = \mathcal{E}$ DEFINED ON $\ell_T^\infty(\mathbb{R}^{n_u})$ | $K_1$ | $0$ | $K_1 r$ |
| $\mathcal{X} = \mathcal{E}$ DEFINED ON $\ell_T^2(\mathbb{R}^{n_u})$ | $K_2$ | $0$ | $K_2 r$ |
| $\mathcal{X} = \{f^{\mathrm{Pool}}\}$ | $1$ | $0$ | $r$ |
| $i$TH SSM BLOCK $\mathcal{X} = \mathcal{F}_i^{\mathrm{DTB}}$ | | | |
| $i = 1$ | $K_2\mu_{\mathcal{F}_1}(K_2 r) + \alpha_1$ | $c_{\mathcal{F}_1}(K_2 r)$ | $\hat{r}_{\mathcal{F}_1}(K_2 r) + \alpha_1 r$ |
| $i > 1$ | $K_1\mu_{\mathcal{F}_i}(K_1 r) + \alpha_i$ | $c_{\mathcal{F}_i}(K_1 r)$ | $\hat{r}_{\mathcal{F}_i}(K_1 r) + \alpha_i r$ |

Table 2: Table of $(\mu_{\mathcal{X}}(r), c_{\mathcal{X}}(r))$ and $\hat{r}_{\mathcal{X}}(r)$ constants for Lemma 5.4. The layer types are each considered component of a deep SSM model as described in Section 4.2. The various constants denoted by some form of $K$ are upper bounds from Assumption 4.10 and $\alpha_i$ are the residual weights. The terms $\mu_{\mathcal{F}_i}(r)$, $c_{\mathcal{F}_i}(r)$ and $\hat{r}_{\mathcal{F}_i}(r)$ are in Table 3 for each type of considered nonlinear layer.

| NONLINEARITY IN $\mathcal{F}_i^{\mathrm{DTB}}$ $\mathcal{X} = \mathcal{F}_i$ | $\mu_{\mathcal{F}_i}(r)$ | $c_{\mathcal{F}_i}(r)$ | $\hat{r}_{\mathcal{F}_i}(r)$ |
|---|---|---|---|
| MLP WITH RELU | $(4K_{W,i})^{M_i+1}$ | $4K_{\mathbf{b},i} \cdot \sum_{q=1}^{M_i}(4K_{W,i})^q$ | $K_{W,i}^{M_i+1}r + K_{\mathbf{b},i} \cdot \sum_{q=1}^{M_i-1} K_{W,i}^q$ |
| MLP WITH SIGMOID | $K_{W,i}^{M_i+1}$ | $(K_{\mathbf{b},i} + 0.5) \cdot \sum_{q=1}^{M_i}(K_{W,i})^q$ | $K_{W,i} + K_{\mathbf{b},i}$ |
| GLU | $16(rz^2 + z)$ $z = K_{\mathrm{GLU},i} + 1$ | $0$ | $r$ |

Table 3: Table of $(\mu_{\mathcal{F}_i}(r), c_{\mathcal{F}_i}(r))$ and $\hat{r}_{\mathcal{F}_i}(r)$ constants for Table 2 and Lemma 5.4.
.

*the ball of radius $\hat{r}_{\mathcal{X}}(r)$, where $\mu_{\mathcal{X}}(r), c_{\mathcal{X}}(r)$ and $\hat{r}_{\mathcal{X}}(r)$ are defined in Table 2 for each SSM blocks and for encoder/decoder layer, i.e. $\mathcal{X} \neq \mathcal{F}_i$, $i \in [L]$ and in Table 3 for the nonlinear layers, i.e. $\mathcal{X} = \mathcal{F}_i$, $i \in [L]$.*

The proof is in Appendix B. The lemma implies that an SSM layer can only increase the input's complexity by the factor $\|\Sigma\|_p$, $p = 1, 2$, and the latter gets smaller as the system gets more stable. The results on the MLP layers rely on proof techniques from Truong (2022b;a) used to bound their Rademacher complexity. These bounds on MLP layers are considered conservative, however improving existing bounds on the Rademacher complexity of MLPs are out of the scope of this paper. The proof for the GLU layer is similar to that of the MLP layer, although handling the elementwise product in GLU requires some additional steps. In contrast to other layers, for GLU the values of $\mu, c$ depend on the magnitude of the inputs. Note that for all the considered layer types except GLU, the $(\mu, c)$-RC property holds for unbounded domains. The only reason we consider bounded domains and ranges is for Lemma 5.4 to hold for GLU layers. This can be seen from the fact that for SSM blocks with MLP nonlinearities, the constants $\mu_{\mathcal{F}_i}(r)$, $c_{\mathcal{F}_i}(r)$ and $\hat{r}_{\mathcal{F}_i}(r)$ are independent of $r$ for all considered $i$, according to Table 3.

To simplify the notation, in the rest of the paper we denote $\mu_{\mathcal{X}}(r), c_{\mathcal{X}}(r)$ and $\hat{r}_{\mathcal{X}}(r)$ by $\mu(r), c(r)$ and $\hat{r}(r)$ respectively, whenever $\mathcal{X}$ is clear from the context, and for $i \in [L]$, we sometimes denote $\mu_{\mathcal{F}_i}(r), c_{\mathcal{F}_i}(r), \hat{r}_{\mathcal{F}_i}(r)$ by $\mu_i(r), c_i(r)$ and $\hat{r}_i(r)$ respectively.

Using Lemma 5.4 and Lemma 5.3 and classical Rademacher complexity based PAC bounds, e.g. see Shalev-Shwartz & Ben-David (2014, Theorem 26.5), leads to a PAC bound for deep SSMs, summarized in the main theorem below.

**Theorem 5.5** (Main). *Let Assumption 4.10 hold. Then for any $\delta \in (0,1)$ we have*

$$\mathbb{P}_{S \sim \mathcal{D}^N} \left( \forall f \in \mathcal{F}: \quad \mathcal{L}(f) - \mathcal{L}_{emp}^S(f) \leq \frac{\mu K_{\mathbf{u}} L_l + c L_l}{\sqrt{N}} + 4 K_l \sqrt{\frac{2 \log\left(\frac{4}{\delta}\right)}{N}} \right) \geq 1 - \delta \tag{12}$$

*where $K_l = 2 L_l \max\{K_{\mathrm{Dec}} r_{L+1}, K_y\}$. The term $r_{L+1}$ is obtained recursively for all $i \in [L+1]$,*

$$r_i = \begin{cases} K_{\mathrm{Enc}} K_{\mathbf{u}} & i = 1 \\ \hat{r}_{\mathcal{F}_1}(K_2 r_1) + \alpha_1 r_1 & i = 2 \\ \hat{r}_{\mathcal{F}_{i-1}}(K_1 r_{i-1}) + \alpha_{i-1} r_{i-1} & i > 2 \end{cases} \tag{13}$$

*where $\hat{r}_{\mathcal{F}_i}(r)$ are as in Table 3 of Lemma 5.4. Moreover, let us define $\mu_1 = \mu_{\mathcal{F}_1}(K_2 r_1)$, $c_1 = c_{\mathcal{F}_1}(K_2 r_1)$, and for $i = 2, \ldots, L$, let us define $\mu_i = \mu_{\mathcal{F}_i}(K_1 r_i)$, $c_i = c_{\mathcal{F}_i}(K_1 r_i)$. Using these quantities, the constants $\mu$ and $c$ are defined as follows*

$$\mu = K_{\mathrm{Enc}} K_{\mathrm{Dec}} (\mu_1 K_2 + \alpha_1) \prod_{i=2}^{L} (\mu_i K_1 + \alpha_i)$$

$$c = K_{\mathrm{Dec}} \left( c_L + \sum_{j=1}^{L-1} \left[ \prod_{i=j+1}^{L} (\mu_i K_1 + \alpha_i) \right] c_j \right). \tag{14}$$

*Sketch of the proof.* From standard PAC bounds involving Rademacher complexity (Shalev-Shwartz & Ben-David, 2014, Theorem 26.5) and the Contraction Lemma (Shalev-Shwartz & Ben-David, 2014, Lemma 26.9), it follows that with probability at least $1-\delta$, for any $f \in \mathcal{F}$, $\mathcal{L}(f) - \mathcal{L}_{emp}^S(f) \leq \mathbb{E}_{\boldsymbol{\sigma}} \left[ \sup_{f \in \mathcal{F}} \left\| \frac{1}{N} \sum_{i=1}^{N} \sigma_i f(\mathbf{u}_i) \right\|_{\ell_T^2(\mathbb{R}^{n_{\mathrm{in}}})} \right] + K_l \sqrt{\frac{2 \log(4/\delta)}{N}}$. From Lemma 5.2 and Lemma 5.4, it follows that the restriction of the elements of $\mathcal{F}$ to the ball $B_{\ell_T^2(\mathbb{R}^{n_{\mathrm{in}}})}(K_{\mathbf{u}})$ of radius $K_{\mathbf{u}}$ in $\ell_T^2(\mathbb{R}^{n_{\mathrm{in}}})$ is $(\mu, c)$-RC with $\mu$ and $c$ as in the statement of the Theorem. Hence, $\mathbb{E}_{\boldsymbol{\sigma}} \left[ \sup_{f \in \mathcal{F}} \left\| \frac{1}{N} \sum_{i=1}^{N} \sigma_i f(\mathbf{u}_i) \right\|_{\ell_T^2(\mathbb{R}^{n_{\mathrm{in}}})} \right] \leq \mu \mathbb{E}_{\boldsymbol{\sigma}} \left[ \left\| \frac{1}{N} \sum_{i=1}^{N} \sigma_i \mathbf{u}_i \right\|_{\ell_T^2(\mathbb{R}^{n_{\mathrm{in}}})} \right] + \frac{c}{\sqrt{N}} \leq \frac{\mu K_{\mathbf{u}} + c}{\sqrt{N}}$. The complete proof can be found in Appendix B. $\square$

**Discussion and interpretation of Theorem 5.5.** The bound vanishes as $N$ grows and remains independent of the sequence length and state dimension — an advantage over typical sequential model bounds that diverge with $T$.

The key constants are $\mu$ and $c$. Note, that for deep SSMs with no MLP layers, $c$ is zero. Intuitively, $\mu$ and $c$ are somewhat analogous to Lipschitz constants of deep networks, although the formal relationship between the two requires future work.

The bound appears exponential in depth, like in case of deep neural networks (Truong, 2022b; Golowich et al., 2018), and includes MLP bounds when MLP layers are used (Golowich et al., 2020; Truong, 2022b). However, SSM layer norms can mitigate this effect: if SSMs are contractions, they counteract the higher Rademacher complexity of nonlinear layers. SSM norms depend on stability — more stable SSMs typically have smaller norms if $C$ and $B$ remain unchanged, suggesting that stability may offset depth effects in both SSM and MLP layers. Moreover, using GLU instead of MLP layers may reduce generalization gaps, as deep SSMs with GLU layers have $c = 0$.

As discussed in Section 4.3, for Theorem 5.5 we only require that the state matrices are Schur, which does not imply weights of the SSM layers are bounded. This intuitively suggests that regularization techniques directly penalizing the norm of the weights of the SSM layers exclude a set of potentially well-generalizing solutions for which the norm of the weights are further from the origin, while the system norm stays low.

*Remark* 5.6 (Proof techniques from prior work on simple RNNs or single layer SSMs.). Prior work establishing PAC bounds on sequential models employ proof techniques that are not applicable to deep SSMs considered in the paper at hand. Comparing our proof to every relevant paper is not feasible, hence we only discuss the most common techniques.

First of all, we are not aware of any prior paper generalization bounds that hold for deep SSMs. It is not clear how to formulate deep SSMs as simple RNNs or Transformers, therefore proof techniques that hold for those models do not automatically hold for deep SSMs.

Many proofs on simple RNNs, Transformers or single layer SSMs (single LTI systems) bound the Rademacher complexity by first bounding covering numbers which serve as natural upper bounds on the Rademacher complexity. However, it is not clear how to bound covering numbers of deep SSMs, even if the covering numbers of the individual layers are known, as the behavior of covering numbers under model composition is not straightforward. One option for dealing with this problem is to use the contraction lemma for covering numbers (Shalev-Shwartz & Ben-David, 2014), or alternatively to apply Talagrand's contraction lemma (Ledoux & Talagrand, 1991) and express the Rademacher complexity of the composition via that of the first layer, followed by covering number arguments. This technique cannot be applied directly to SSM blocks due to the nonlinearity being parametrized. Even if this was possible, the Lipschitz constant of such parametrized nonlinearity is often hard to estimate.

The required assumptions for PAC bounds on the generalization error of simple RNNs, single layer SSMs or Transformers usually employ an explicit upper bound on the parameter norm of the considered model. This is often the case when proofs based on covering numbers are used. This is a stronger assumption than what we require as it is possible to define SSM layers with arbitrarily large parameter norm, but which are stable with uniformly bounded $H_2$ or $\ell_1$ system norms.

Finally, majority of the generalization bounds for similar model structures often depend (exponentially) on the sequence length $T$ and only hold for a single time-mixing layer.

*Remark* 5.7 (The role of Rademacher Contractions in the proof of Theorem 5.5.). The use of RC allows us to overcome all the obstacles from Remark 5.6. Namely, by Lemma 5.2-5.4, it is possible to upper bound the Rademacher complexity of an arbitrarily deep model whose components satisfy the $(\mu, c)$-RC inequality. As $(\mu, c)$-RC is defined for mappings between arbitrary Banach-spaces, therefore it is a generalization of Talagrand's lemma and can be applied to time-mixing model components which act on Banach-spaces of sequences and which contain components with parametrized nonlinearities without explicit Lipschitz constraints. Additionally, $(\mu, c)$-RC allows us to establish a time-independent bound for deep SSMs with stable LTI layers due to the $\mu$ and $c$ terms of such model components not depending on $T$.

We remark that the 4th point in Assumption 4.10 requiring a bound on the $\ell^2$ norms of input sequences is also necessary (but not sufficient) for deriving a time-independent bound. Without this assumption the bound would depend linearly on $T$, if the stability assumption still holds. If we drop the stability assumption, then our proof technique may still be used, but it will result in a bound with a possibly exponential dependence on $T$.

# 6 Numerical example

In order to illustrate our results, we trained a model consisting of a single SSM layer on a binary classification problem of separating the elements of two intertwined spirals, with a training set containing $N$ sequences of length $T$, for various values of $N$. Let $\{\theta_t\}_{1 \leq t \leq T}$ be a standard normal sample sorted in ascending order and $\varphi_t = 7\pi\sqrt{\theta_t}$. Consider the following two classes of time series, labeled by 0 and 1.

$$\mathbf{x}_0[t] = ((2\varphi_t + \pi)\cos(\varphi_t), (2\varphi_t + \pi)\sin(\varphi_t))^T + \boldsymbol{\varepsilon}$$
$$\mathbf{x}_1[t] = (-(2\varphi_t + \pi)\cos(\varphi_t), -(2\varphi_t + \pi)\sin(\varphi_t))^T + \boldsymbol{\varepsilon}$$

where $1 \leq t \leq T$ for some $T$ and $\boldsymbol{\varepsilon}$ are i.i.d standard normal noise vectors. The training data is depicted on Figure 1.

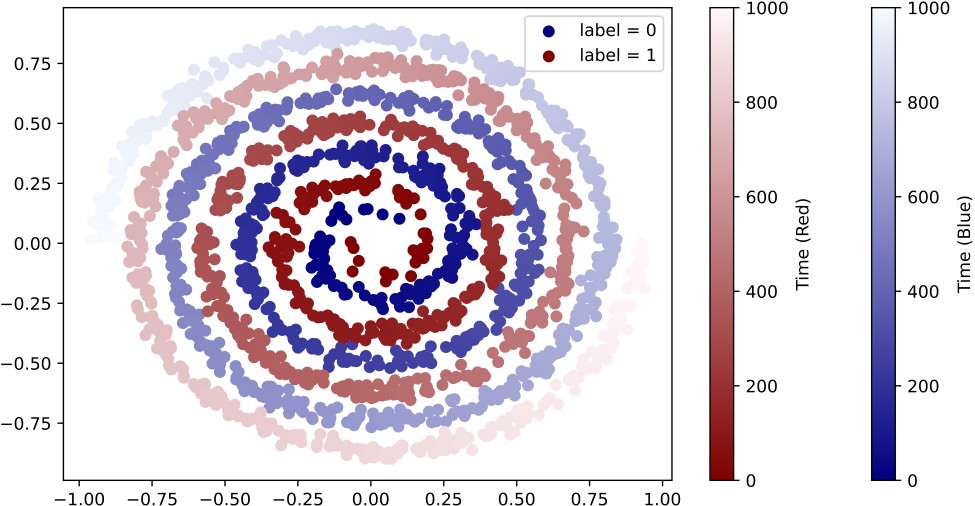

Figure 1: Dataset containing two classes of spiral curves. This dataset is used in our experiment to illustrate our result

.

We trained a linear SSM on this binary classification problem with a training set size of $N$, for various values of $N$. The resulting parameter vector therefore depends on $N$. We applied the Adam optimizer to the binary cross-entropy loss combined with applying a sigmoid activation to the scores outputted by the model. Theorem 5.5 states that with high probability, we can upper bound the true loss with the sum of the empirical loss and a bounding term.

Namely, let $f^N$ be the model obtained by preforming the training on an $N$-sized training set. Then by Theorem 5.5, with probability at least $1 - \delta$ over the choice of $S$ we have that for all $N \in \mathbb{N}$

$$\mathcal{L}(f^N) \leq \underbrace{\mathcal{L}^S_{emp}(f^N) + \frac{K_{\mathcal{F}} + K_l\sqrt{2\log(\frac{4}{\delta})}}{\sqrt{N}}}_{=:r(N,\delta)} \tag{15}$$

Consequently, we can plot the left and right hand side of inequality 15 for many values of $N$ and expect the curve $N \mapsto r(N,\delta) = \mathcal{L}^S_{emp}(f^N) + \frac{K_{\mathcal{F}}+K_l\sqrt{2\log(\frac{4}{\delta})}}{\sqrt{N}}$ to be roughly shaped as $\mathcal{L}(f^N) + \frac{c}{\sqrt{N}}$ for a constant $c > 0$ and for large enough $N$. Moreover, the curve $N \mapsto \mathcal{L}(f^N)$ should be under the curve of $N \mapsto r(N,\delta)$. Moreover, we expect that for some values of $N_1$ and $N_2$, $\mathcal{L}(f^{N_1}) > r(N_2,\delta)$ holds, at least for some model classes and some datasets. Otherwise, the bound would be vacuous and thus not very useful. An example of a vacuous bound would be one where $r(N,\delta) \geq 1$ and the loss function is between $[0, 1]$. In this case the bound would say that the true loss is smaller than 1, which, while true, is not very useful. We show that for the example at hand, there exist $N_1, N_2 \in \mathbb{N}$ such that $\mathcal{L}(f^{N_1}) > r(N_2,\delta)$, and hence the bound is non-vacuous.

In this case, the true loss is estimated by taking the loss of the model on a very large set of samples (concretely, 150 000). It can be seen on Figure 2 that the bound holds and in this scenario it is non-vacuous, i.e. there exists a value of the true loss - highlighted by the red broken line - which is greater, than the value of the estimation at a different value of $N$.

Moreover, it can also be seen on Figure 3 that during the learning process, the numerical value of the bounding term, and therefore the estimation of the true loss, correlates with the true performance of the model. Once the learning algorithm passed its optimum where the accuracy is close to 1, and starts to exhibit overfitting, the bounding term and the estimation start to rapidly grow, while the value of the loss stays consistently low. This suggests that adding the bounding term to the loss function as a regularization term during training could be beneficial. We consider this as a research topic on its own and out of the scope of this paper.

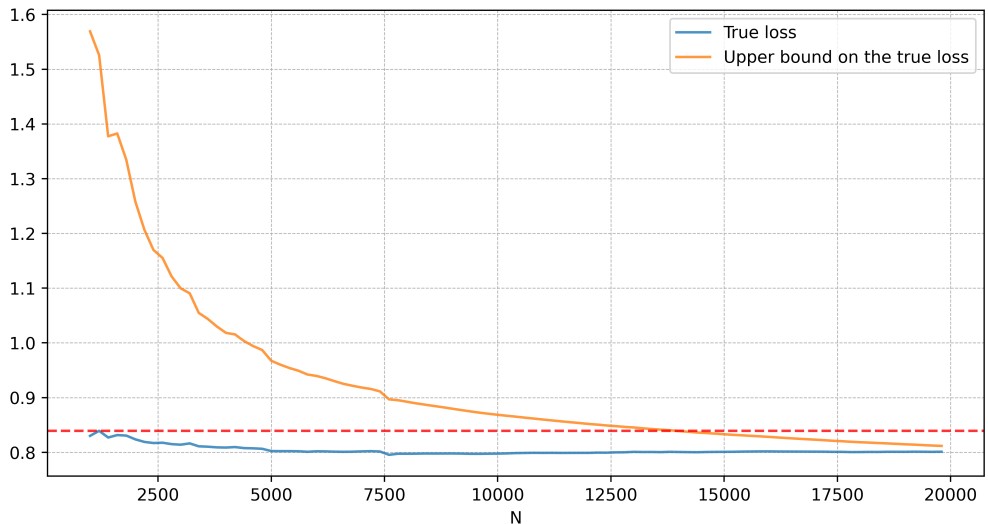

Figure 2: Upper bound on the true loss by taking the empirical loss and the bounding term from Theorem 5.5 for various values of $N$. This is standard figure in PAC literature and illustrate the behavior of the generalization bound in Theorem 5.5. The orange curve is the proposed upper bound $r(N, \delta) = \mathcal{L}^S_{emp}(f^N) + \frac{K_\mathcal{F} + K_l \sqrt{2 \log(\frac{4}{\delta})}}{\sqrt{N}}$, and the blue curve is the true loss $\mathcal{L}(f^N)$ of the learned model $f^N$. The model $f^N$ was learned from a training dataset with $N$ data points. The true loss $\mathcal{L}(f^N)$ is approximated by the empirical loss on a validation dataset with 150 000 data points. The fact that the orange curve is above the blue one illustrates that the statement of Theorem 5.5 indeed holds, i.e., $\mathcal{L}(f^N) \leq r(N, \delta)$. The orange curve converges to the blue one at rate $O(\frac{1}{\sqrt{N}})$, i.e., for a constant $c$ and for large enough $N$, the bound $r(N, \delta)$ is roughly $\mathcal{L}(f^N) + \frac{c}{\sqrt{N}}$. The fact that the orange line ($r(N, \delta)$) descends below the red dashed line (the maximal value $\max_N \mathcal{L}(f^N)$ of the true loss) indicates that the bound $r(N_1, \delta)$ for some $N_1$ ($N_1$ is around 15 000) is strictly smaller than the true loss at $\mathcal{L}(f^{N_2})$ for some $N_2$ ($N_2$ is around 1250), i.e., $\mathcal{L}(N_2) > r(N_1, \delta)$, meaning the bound is non-vacuous in this scenario.

# 7 Conclusions

We derive generalization bounds for deep SSMs by decomposing the architecture into components satisfying the definition of *Rademacher Contraction*. Under reasonable stability conditions, the bound is sequence-length independent and improves on prior results for linear RNNs as those bounds depend on the sequence-length exponentially. Given that stability is central to state-of-the-art SSMs (e.g., S4, S5, LRU), our work offers insight into their strong long-range performance.

We introduce the concept of *Rademacher Contraction* which we believe is a powerful tool for determining the complexity of a wide variety of stacked architectures including feedforward and recurrent elements.

Our contraction-based approach, while reasonable, may yield conservative bounds for complex SSMs or deep stacks. However, with state-of-the-art SSMs using fewer than ten blocks, this is a minor concern. A key limitation is that all elements must be *Rademacher Contractions*, which may be too restrictive for complex functions.

Future work includes extending these results to learning from limited (possibly single) time-series and deriving tighter bounds, potentially using concentration inequalities for mixing processes and the PAC-Bayesian framework. Incorporating Mamba-like architectures into our framework can also be a subject of future research.

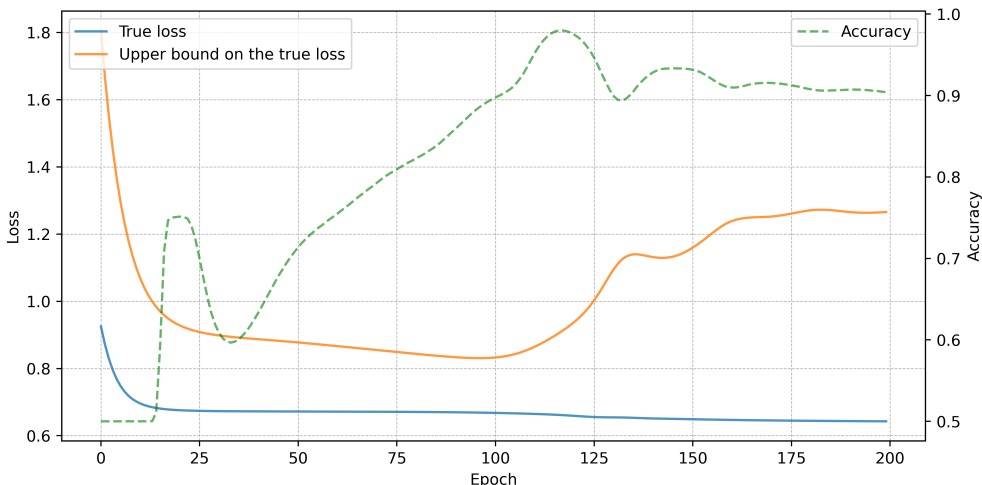

Figure 3: Behavior of the bound on the true loss during learning. The proposed PAC bound $r(N, \delta) = \mathcal{L}_{emp}^S(f^N) + \frac{K_\mathcal{F} + K_l \sqrt{2 \log(\frac{4}{\delta})}}{\sqrt{N}}$ (orange line) is always greater than the actual true loss $\mathcal{L}(f^N)$ (blue line) during learning, a simple consequence of Theorem 5.5. Moreover, the behavior of the bound strongly correlates with the model's true performance in accuracy (green dashed line), where by accuracy we mean the complement $1 - \mathbf{E}_{(\mathbf{u},y)\sim\mathcal{D}}[\ell_{0-1}(f^N(\mathbf{u}), y)]$ of the true loss $\mathbf{E}_{(\mathbf{u},y)\sim\mathcal{D}}[\ell_{0-1}(f^N(\mathbf{u}), y)]$ for the $0-1$ binary loss function $\ell_{0-1}(y, y') = \begin{cases} 1 & y \neq y' \\ 0 & y = y' \end{cases}$ . This true loss was approximated by the corresponding empirical loss on a validation dataset with 150 000 data points. Recall that our model is a classifier and we used the cross-entropy loss function instead of the $0-1$ binary loss for defining true and empirical losses for training and applying Theorem 5.5. The bound $r(N, \delta)$ stays low up to the point where the accuracy starts to decrease, where the bound starts to grow rapidly.

## Acknowledgements

This research was supported in part by the European Union project RRF-2.3.1-21-2022-00004 within the Artificial Intelligence National Laboratory (MILAB); in part by the C.N.R.S. E.A.I. Project "Stabilité des algorithmes d'apprentissage pour les réseaux de neurones profonds et récurrents en utilisant la géométrie et la théorie du contrôle via la compréhension du rôle de la surparamétrisation (StabLearnDyn)"; and in part by the E.D.F. project FaRADAI under Grant 101103386.

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

## A  Lipschitzness of the cross-entropy loss

For simplicity, we consider the binary cross-entropy loss for the scalar output case as an elementwise loss function, defined as

$$\ell(x, y) = -y \log(x) - (1 - y) \log(1 - x). \tag{16}$$

The function defined in equation 16 without any other assumptions is not Lipschitz in the sense of Assumption 4.10 on the $[0, 1]$ interval. However, if we bound the argument of the cross-entropy away from 0, i.e. it is defined on $[a, 1]$ for some positive $a$, it is Lipschitz in the sense of Assumption 4.10 with a Lipschitz constant proportional to $a^{-1}$.

A more practical assumption is that the cross-entropy is combined with the softmax function. For the scalar output case, we have

$$\ell(x, y) = -y \log(\text{sigmoid}(x)) - (1 - y) \log(1 - \text{sigmoid}(x)) \tag{17}$$

for $y \in [0, 1]$. If $x$ belongs to the interval $[-a, a]$, i.e., the model outputs are bounded, we have

$$\left| \frac{\partial \ell}{\partial x} \right| \leq 2 \quad \text{and} \quad \left| \frac{\partial \ell}{\partial y} \right| \leq a$$

hence the function defined in equation 17 is 2-Lipschitz in $x$ and $a$-Lipschitz in $y$ and it is $\max\{2, a\}$-Lipschitz as long as $y \in [0, 1]$. We can extend $\ell$ to all $y$ by setting $\ell(x, y) = \ell(x, 0)$ for $y < 0$ and $\ell(x, y) = \ell(x, 1)$ for $y > 1$, without changing the Lipschitz constant.

This argument holds for the case when the model outputs a vector and we apply softmax, in a straightforward manner. We omit the vector output case from the paper, because it makes the proof more technical and less readable, while all the key terms remain the same. For the vector output case, in the first half of the proof of Theorem 5.5 in Appendix B, instead of applying the Contraction Lemma for the Rademacher complexity, we need to apply Maurer (2016, Corollary 4).

For cross-entropy with softmax, we can also consider the labels built into the loss function and investigate the Lipschitzness in the input variable only, i.e. instead of $\ell(x, y)$ we have $\ell(x)$. This definition differs from what we have in the main text, however, our proof and the above argument works, while the Lipschitz constant is at most 2 without depending on $a$ in this case.

## B  Proofs

In this section we need to prove $(\mu, c)$-RC property for linear (or affine) transformations which are constant in time, in many cases. For better readability, we only do the calculations once and use it as a lemma.

**Lemma B.1.** *Let $\mathcal{X}_1, \mathcal{X}_2$ be two Banach spaces with norms $\| \cdot \|_{\mathcal{X}_1}$ and $\| \cdot \|_{\mathcal{X}_2}$, and for every bounded linear operation $W : \mathcal{X}_1 \to \mathcal{X}_2$ and any $\underline{\mathbf{b}} \in \mathcal{X}_2$ $f_{W, \underline{\mathbf{b}}}(\mathbf{u}) = W(\mathbf{u}) + \underline{\mathbf{b}} \in \mathcal{X}_2$. Let us denote by $\|W\|_{op}$ the induced norm of a bounded linear operator $W : \mathcal{X}_1 \to \mathcal{X}_2$, i.e., $\|W\|_{\text{op}} := \sup_{x \in \mathcal{X}_1} \frac{\|W(x)\|_{\mathcal{X}_2}}{\|x\|_{\mathcal{X}_1}}$. Let us assume that $W \in \mathcal{W}$ such that $\sup_{W \in \mathcal{W}} \|W\|_{\text{op}} < K_W$ and $\underline{\mathbf{b}} \in \mathcal{B}$ such that $\sup_{\underline{\mathbf{b}} \in \mathcal{B}} \|\underline{\mathbf{b}}\|_{\mathcal{X}_2} < K_{\mathbf{b}}$. Then the set of transformations $\mathcal{F} = \{f_{W, \underline{\mathbf{b}}} \mid W \in \mathcal{W}, \underline{\mathbf{b}} \in \mathcal{B}\}$ is $(K_W, K_{\mathbf{b}})$-RC, and the image of the ball $B_{\mathcal{X}_1}(r)$ under $f \in \mathcal{F}$ is contained in $B_{\mathcal{X}_2}(K_W r + K_{\mathbf{b}})$.*

*Remark* B.2. *We are mainly interested in the cases when $\mathcal{X}_1 = \ell_T^q(\mathbb{R}^{n_u})$, and $\mathcal{X}_1 = \ell_T^l(\mathbb{R}^{n_y})$, $(q, l) \in \{(2, 2), (2, \infty), (\infty, \infty)\}$. For the special case of affine transformations that are constant in time, i.e. $f(\mathbf{u})[k] = W\mathbf{u}[k] + \mathbf{b}$ for a weight matrix $W \in \mathbb{R}^{n_v \times n_u}$ and bias term $\mathbf{b} \in \mathbb{R}^{n_v}$ for all $k \in [T]$, the operator norm equals the corresponding matrix norm, i.e $\|W\|_{\text{op}} = \|W\|_{q, \infty}$. In this case, $\underline{\mathbf{b}}$ is the sequence for which $\underline{\mathbf{b}}[k] = \mathbf{b}$ for all $k \in [T]$, thus $\|\underline{\mathbf{b}}\|_{\ell_T^\infty(\mathbb{R}^{n_v})} = \|\mathbf{b}\|_\infty$.*

*Proof.* First, let us prove a simple fact about Rademacher random variables that we will need, namely if $\boldsymbol{\sigma} = (\sigma_1, \ldots, \sigma_N)^T$ and $\sigma_i$ are i.i.d. Rademacher variables, then

$$\mathbb{E}_{\boldsymbol{\sigma}}\left[\left|\sum_{i=1}^{N}\sigma_i\right|\right] \leq \sqrt{N}. \tag{18}$$

This is true, because

$$\mathbb{E}_{\boldsymbol{\sigma}}\left[\left|\sum_{i=1}^{N}\sigma_i\right|\right] = \sqrt{\left(\mathbb{E}_{\boldsymbol{\sigma}}\left[\left|\sum_{i=1}^{N}\sigma_i\right|\right]\right)^2} \leq \sqrt{\mathbb{E}_{\boldsymbol{\sigma}}\left[\left|\sum_{i=1}^{N}\sigma_i\right|^2\right]}$$

$$= \sqrt{\mathbb{E}_{\boldsymbol{\sigma}}\left[\sum_{i=1}^{N}\sigma_i^2 + 2\sum_{i,j=1}^{N}\sigma_i\sigma_j\right]} = \sqrt{\sum_{i=1}^{N}\mathbb{E}_{\boldsymbol{\sigma}}\left[\sigma_i^2\right] + 2\sum_{i,j=1}^{N}\mathbb{E}_{\boldsymbol{\sigma}}\left[\sigma_i\sigma_j\right]} = \sqrt{N},$$

where the first inequality follows from Jensen's inequality and the last equality follows from the linearity of the expectation, and the facts that $\sigma_i$ are Rademacher variables and form and i.i.d sample.

For $Z \in \mathcal{X}_1$ we have

$$\mathbb{E}_{\boldsymbol{\sigma}}\left[\sup_{(W,\underline{\mathbf{b}})\in\mathcal{W}\times\mathcal{B}}\sup_{\{\mathbf{u}_i\}_{i=1}^N\in Z}\left\|\frac{1}{N}\sum_{i=1}^{N}\sigma_i(W(\mathbf{u}_i)+\underline{\mathbf{b}})\right\|_{\mathcal{X}_2}\right]$$

$$\leq \mathbb{E}_{\boldsymbol{\sigma}}\left[\sup_{W\in\mathcal{W}}\sup_{\{\mathbf{u}_i\}_{i=1}^N\in Z}\left\|\frac{1}{N}\sum_{i=1}^{N}\sigma_i W(\mathbf{u}_i)\right\|_{\mathcal{X}_2}\right] + \mathbb{E}_{\boldsymbol{\sigma}}\left[\sup_{\underline{\mathbf{b}}\in\mathcal{B}}\left\|\frac{1}{N}\sum_{i=1}^{N}\sigma_i\underline{\mathbf{b}}\right\|_{\mathcal{X}_2}\right]$$

$$= \mathbb{E}_{\boldsymbol{\sigma}}\left[\sup_{W\in\mathcal{W}}\sup_{\{\mathbf{u}_i\}_{i=1}^N\in Z}\left\|W\left(\frac{1}{N}\sum_{i=1}^{N}\sigma_i\mathbf{u}_i\right)\right\|_{\mathcal{X}_2}\right] + \mathbb{E}_{\boldsymbol{\sigma}}\left[\sup_{\underline{\mathbf{b}}\in\mathcal{B}}\left\|\frac{1}{N}\sum_{i=1}^{N}\sigma_i\underline{\mathbf{b}}\right\|_{\mathcal{X}_1}\right]$$

$$\leq \mathbb{E}_{\boldsymbol{\sigma}}\left[\sup_{W\in\mathcal{W}}\|W\|_{\mathrm{op}}\sup_{\{\mathbf{u}_i\}_{i=1}^N\in Z}\left\|\frac{1}{N}\sum_{i=1}^{N}\sigma_i\mathbf{u}_i\right\|_{\mathcal{X}_1}\right] + \mathbb{E}_{\boldsymbol{\sigma}}\left[\frac{1}{N}\left|\sum_{i=1}^{N}\sigma_i\right|\sup_{\underline{\mathbf{b}}\in\mathcal{B}}\|\underline{\mathbf{b}}\|_{\mathcal{X}_2}\right]$$

$$\leq \sup_{W\in\mathcal{W}}\|W\|_{\mathrm{op}}\mathbb{E}_{\boldsymbol{\sigma}}\left[\sup_{\{\mathbf{u}_i\}_{i=1}^N\in Z}\left\|\frac{1}{N}\sum_{i=1}^{N}\sigma_i\mathbf{u}_i\right\|_{\mathcal{X}_1}\right] + \sup_{\underline{\mathbf{b}}\in\mathcal{B}}\|\underline{\mathbf{b}}\|_{\mathcal{X}_2}\mathbb{E}_{\boldsymbol{\sigma}}\left[\frac{1}{N}\left|\sum_{i=1}^{N}\sigma_i\right|\right]$$

$$\leq \sup_{W\in\mathcal{W}}\|W\|_{\mathrm{op}}\mathbb{E}_{\boldsymbol{\sigma}}\left[\sup_{\{\mathbf{u}_i\}_{i=1}^N\in Z}\left\|\frac{1}{N}\sum_{i=1}^{N}\sigma_i\mathbf{u}_i\right\|_{\mathcal{X}_1}\right] + \frac{1}{\sqrt{N}}\sup_{\underline{\mathbf{b}}\in\mathcal{B}}\|\underline{\mathbf{b}}\|_{\mathcal{X}_2}$$

where the first inequality follows from the triangle inequality, the first equality is the linearity of $W$, the second inequality follows from the definition of the operator norm, while the third and fourth inequalities refer only to the bias term and follow from the absolute homogeneity of the norm and inequality 18.

We can see that the calculations hold if the transformations are restricted to the ball $B_{\mathcal{X}_1}(r)$ for any choice of $\mathcal{X}_1$ we consider. The radius can grow as

$$\|W(\mathbf{u})+\underline{\mathbf{b}}\|_{\mathcal{X}_2} \leq \|W(\mathbf{u})\|_{\mathcal{X}_2} + \|\underline{\mathbf{b}}\|_{\mathcal{X}_2} \leq \|W\|_{\mathrm{op}}\|\mathbf{u}\|_{\mathcal{X}_1} + \|\underline{\mathbf{b}}\|_{\mathcal{X}_2}.$$

Remark B.2 is straightforward from the definitions of the considered norms. $\square$

*Proof of Lemma 5.2.* Let the Banach spaces which contain $X_i$ be denoted by $\mathcal{X}_i$ for $i = 1, 2, 3$. Let $Z \subseteq X_1^N$ and $\tilde{Z} = \{\{\varphi_1(\mathbf{u}_i)\}_{i=1}^N \mid \varphi_1 \in \Phi_1\}$. We have

$$\mathbb{E}_{\boldsymbol{\sigma}}\left[\sup_{\varphi_2\in\Phi_2}\sup_{\varphi_1\in\Phi_1}\sup_{\{\mathbf{u}_i\}_{i=1}^N\in Z}\left\|\frac{1}{N}\sum_{i=1}^{N}\sigma_i\varphi_2(\varphi_1(\mathbf{u}_i))\right\|_{\mathcal{X}_3}\right]$$

$$
= \mathbb{E}_{\boldsymbol{\sigma}} \left[ \sup_{\varphi_2 \in \Phi_2} \sup_{\{\mathbf{v}_i\}_{i=1}^N \in \tilde{Z}} \left\| \frac{1}{N} \sum_{i=1}^N \sigma_i \varphi_2(\mathbf{v}_i) \right\|_{\mathcal{X}_3} \right]
$$

$$
\leq \mu_2 \mathbb{E}_{\boldsymbol{\sigma}} \left[ \sup_{\varphi_1 \in \Phi_1} \sup_{\{\mathbf{u}_i\}_{i=1}^N \in Z} \left\| \frac{1}{N} \sum_{i=1}^N \sigma_i \varphi_1(\mathbf{u}_i) \right\|_{\mathcal{X}_2} \right] + \frac{c_2}{\sqrt{N}}
$$

$$
\leq \mu_2 \mu_1 \mathbb{E}_{\boldsymbol{\sigma}} \left[ \sup_{\{\mathbf{u}_i\}_{i=1}^N \in Z} \left\| \frac{1}{N} \sum_{i=1}^N \sigma_i \mathbf{u}_i \right\|_{\mathcal{X}_1} \right] + \mu_2 \frac{c_1}{\sqrt{N}} + \frac{c_2}{\sqrt{N}}
$$

$\square$

*Proof of Lemma 5.4.* **Encoder and decoder.** The encoder is case **a)**, while the decoder is case **b)** in Lemma B.1 along with Remark B.2.

**SSM.** As discussed in Section 4.2, an SSM is equivalent to a linear transformation called its input-output map. Therefore, by Lemma B.1, the SSM is $(\mu, 0)$-RC in both cases, where $\mu$ is the operator norm of the input-output map. Combining this with Lemma 4.3 yields the result.

*Remark* B.3. As the value of $T$ is fixed, the input-output map can be described by the so-called Toeplitz matrix of the system. In this case, the operator norm equals to the appropriate induced matrix norm of the Toeplitz matrix. For the case of $T = \infty$, the input-output map still exists and is a linear operator. The proof of Lemma B.1 holds in this case as well for operator norms.

**Pooling.** For any $Z \subseteq \ell_T^\infty(\mathbb{R}^{n_u})$ we have

$$
\mathbb{E}_{\boldsymbol{\sigma}} \left[ \sup_{\{\mathbf{z}_i\}_{i=1}^N \in Z} \left\| \frac{1}{N} \sum_{i=1}^N \sigma_i f^{\text{Pool}}(\mathbf{z}_i) \right\|_\infty \right]
$$

$$
= \mathbb{E}_{\boldsymbol{\sigma}} \left[ \sup_{\{\mathbf{z}_i\}_{i=1}^N \in Z} \sup_{1 \leq j \leq n_u} \left| \frac{1}{N} \sum_{i=1}^N \sigma_i \left( \frac{1}{T} \sum_{k=1}^T \mathbf{z}_i^{(j)}[k] \right) \right| \right]
$$

$$
= \mathbb{E}_{\boldsymbol{\sigma}} \left[ \sup_{\{\mathbf{z}_i\}_{i=1}^N \in Z} \sup_{1 \leq j \leq n_u} \left| \frac{1}{T} \sum_{k=1}^T \left( \frac{1}{N} \sum_{i=1}^N \sigma_i \mathbf{z}_i^{(j)}[k] \right) \right| \right]
$$

$$
\leq \mathbb{E}_{\boldsymbol{\sigma}} \left[ \sup_{\{\mathbf{z}_i\}_{i=1}^N \in Z} \frac{1}{T} \sum_{k=1}^T \sup_{1 \leq j \leq n_u} \left| \frac{1}{N} \sum_{i=1}^N \sigma_i \mathbf{z}_i^{(j)}[k] \right| \right]
$$

$$
= \mathbb{E}_{\boldsymbol{\sigma}} \left[ \sup_{\{\mathbf{z}_i\}_{i=1}^N \in Z} \frac{1}{T} \sum_{k=1}^T \left\| \frac{1}{N} \sum_{i=1}^N \sigma_i \mathbf{z}_i[k] \right\|_\infty \right]
$$

$$
\leq \mathbb{E}_{\boldsymbol{\sigma}} \left[ \sup_{\{\mathbf{z}_i\}_{i=1}^N \in Z} \left\| \frac{1}{N} \sum_{i=1}^N \sigma_i \mathbf{z}_i \right\|_{\ell_T^\infty(\mathbb{R}^{n_u})} \right]
$$

**MLP.** For both type of activation functions we will prove the result by first proving it for single layer networks. To this end, let $\rho$ be an activation function, which is either ReLU or a sigmoid with the properties stated in Assumption 4.10.

Consider constants $K_W, K_{\mathbf{b}} > 0$ and integers $m, n_v > 0$. We first consider the family $\mathcal{F}_{MLP,K_W,K_{\mathbf{b}},\rho,m,n_v}$ of single hidden layer neural networks $f : \ell_T^\infty(\mathbb{R}^m) \to \ell_T^\infty(\mathbb{R}^{n_v})$ defined by $f(\mathbf{u})[k] = \rho(g(\mathbf{u}[k]))$, where $g(x) = W\mathbf{x} + \mathbf{b}$ is the preactivation function and $g$ belongs to the set $\mathcal{G}_{K_W,K_{\mathbf{b}},m,n_v} = \{g : \mathbf{x} \mapsto W\mathbf{x} + \mathbf{b} \mid W \in \mathcal{W}, \mathbf{b} \in \mathcal{B}\}$, where $\mathcal{W} = \{W \in \mathbb{R}^{n_v \times m} \mid \|W\|_{\infty,\infty} < K_W\}$ and $\mathcal{B} = \{\mathbf{b} \in \mathbb{R}^{n_v} \mid \|\mathbf{b}\|_\infty < K_{\mathbf{b}}\}$.

We will show that $\mathcal{F}_{MLP,K_W,K_\mathbf{b},\rho,m,n_v}$ is $(K_W, K_\mathbf{b} + 0.5)$-RC if $\rho$ is sigmoid, and it is $(4K_W, 4K_\mathbf{b})$-RC if $\rho$ is ReLU. Moreover, the elements of $\mathcal{F}_{MLP,K_W,K_\mathbf{b},\rho,m,n_v}$ map balls or radius $r$ to balls of radius $\hat{r}(r)$, where $\hat{r}(r) = K_W + K_\mathbf{b}$ if $\rho$ is sigmoid, and $\hat{r}(r) = K_W r + K_\mathbf{b}$ if $\rho$ is ReLU.

From this the statement of the lemma can be derived as follows. Let $\mathcal{G}_{K_{W,i},K_{\mathbf{b},i},n_M,n_u}$ be the set of all models $f(\mathbf{u})[k] = g(\mathbf{u}[k])$ such that $g \in G_{K_{W,i},K_{\mathbf{b},i},n_M,n_u}$. Notice that $\mathcal{F}_i$ is contained in the composition (as defined in Lemma 5.2) $\mathcal{G}_{K_{W,i},K_{\mathbf{b},i},n_M,n_u} \circ \mathcal{F}_{K_{W,i},K_{\mathbf{b},i},n_M,n_{M-1},\rho_i} \circ \cdots \circ \mathcal{F}_{K_{W,i},K_{\mathbf{b},i},n_y,n_2,\rho_i}$ for suitable integers $n_j$, $j \in \{2, 3, \ldots, M\}$. From the discussion above, $\mathcal{F}_{K_{W,i},K_{\mathbf{b},i},n_j,n_{j+1},\rho_i}$ is $(K_{W,i}, K_{\mathbf{b},i} + 0.5)$-RC (sigmoid) or $(4K_W, 4K_\mathbf{b})$-RC (ReLU) and its elements map balls of radius $r$ to balls of radius 1 (sigmoid) or $K_W r + K_\mathbf{b}$ (ReLU). From Lemma B.1 and Remark B.2 it follows that that $\mathcal{G}_{K_{W,i},K_{\mathbf{b},i},n_u,n_M}$ is $(K_{W,i}, K_{\mathbf{b},i})$-RC and its element map ball of radius $r$ to balls of radius $K_{W,i} r + K_{\mathbf{b},i}$. The statement of the lemma follows now by repeated application of Lemma 5.2.

It is left to prove the claims for single layer MLPs with sigmoid and ReLU activation functions respectively.

**Single layer MLP with sigmoid activations.** Let $\rho$ be a sigmoid such that it is 1-Lipschitz, $\rho(x) \in [-1, 1]$, $\rho(0) = 0.5$, $\rho(x) - \rho(0)$ is odd. Let $G = \mathcal{G}_{K_W,K_\mathbf{b},m,n_v} = \{g : \mathbf{x} \mapsto W\mathbf{x} + \mathbf{b} \mid W \in \mathcal{W}, \mathbf{b} \in \mathcal{B}\}$. Recall that for an input sequence $\mathbf{z} \in \ell_T^\infty(\mathbb{R}^m)$ and function $g \in G$, $g(\mathbf{z}) \in \ell_T^\infty(\mathbb{R}^{n_v})$ means that we apply $g$ for each timestep independently, i.e. $g(\mathbf{z})[k] = g(\mathbf{z}[k])$. We have

$$\mathbb{E}_{\boldsymbol{\sigma}} \left[ \sup_{g \in \mathcal{G}} \sup_{\{\mathbf{z}_i\}_{i=1}^N \in Z} \left\| \frac{1}{N} \sum_{i=1}^N \sigma_i \rho(g(\mathbf{z}_i)) \right\|_{\ell_T^\infty(\mathbb{R}^{n_u})} \right]$$

$$= \mathbb{E}_{\boldsymbol{\sigma}} \left[ \sup_{(W,\mathbf{b}) \in \mathcal{W} \times \mathcal{B}} \sup_{\{\mathbf{z}_i\}_{i=1}^N \in Z} \sup_{1 \leq k \leq T} \left\| \frac{1}{N} \sum_{i=1}^N \sigma_i \rho(W\mathbf{z}_i[k] + \mathbf{b}) \right\|_\infty \right]$$

Let $\mathbf{x}_i = i$, $i = 1, \ldots, N$ and let $\mathcal{H} = \{h_{W,\mathbf{b},\underline{z},k} \mid (W, \mathbf{b}, \underline{z}, k) \in \mathcal{W} \times \mathcal{B} \times (Z \cup \{0\}) \times [T]\}$ such that $h_{W,\mathbf{b},\underline{z},k}(\mathbf{x}_i) = W(\mathbf{z}_i[k]) + \mathbf{b}$. Under our assumptions $\mathcal{H}$ is symmetric to the origin, meaning that $h \in \mathcal{H}$ implies $-h \in \mathcal{H}$. Indeed, notice that if $(W, \mathbf{b}) \in \mathcal{W} \times \mathcal{B}$ then $(-W, -\mathbf{b}) \in \mathcal{W} \times \mathcal{B}$, and hence $h_{-W,-\mathbf{b},\underline{z},k} = -h_{W,\mathbf{b},\underline{z},k}$ also belongs to $\mathcal{H}$. We can apply Theorem 2 from Truong (2022b) for the sigmoid activation $\rho$ and by using that $\rho(x) - \rho(0)$ is odd, we derive the following.

$$\mathbb{E}_{\boldsymbol{\sigma}} \left[ \sup_{(W,\mathbf{b}) \in \mathcal{W} \times \mathcal{B}} \sup_{\{\mathbf{z}_i\}_{i=1}^N \in Z} \sup_{1 \leq k \leq T} \left\| \frac{1}{N} \sum_{i=1}^N \sigma_i \rho(W\mathbf{z}_i[k] + \mathbf{b}) \right\|_\infty \right]$$

$$= \mathbb{E}_{\boldsymbol{\sigma}} \left[ \sup_{h \in \mathcal{H}} \left\| \frac{1}{N} \sum_{i=1}^N \sigma_i \rho(h(\mathbf{x}_i)) \right\|_\infty \right]$$

$$\leq \mathbb{E}_{\boldsymbol{\sigma}} \left[ \sup_{h \in \mathcal{H}} \left\| \frac{1}{N} \sum_{i=1}^N \sigma_i h(\mathbf{x}_i) \right\|_\infty \right] + \frac{1}{2\sqrt{N}}$$

$$= \mathbb{E}_{\boldsymbol{\sigma}} \left[ \sup_{(W,\mathbf{b}) \in \mathcal{W} \times \mathcal{B}} \sup_{\{\mathbf{z}_i\}_{i=1}^N \in Z} \sup_{1 \leq k \leq T} \left\| \frac{1}{N} \sum_{i=1}^N \sigma_i (W\mathbf{z}_i[k] + \mathbf{b}) \right\|_\infty \right] + \frac{1}{2\sqrt{N}}$$

$$= \mathbb{E}_{\boldsymbol{\sigma}} \left[ \sup_{(W,\mathbf{b}) \in \mathcal{W} \times \mathcal{B}} \sup_{\{\mathbf{z}_i\}_{i=1}^N \in Z} \left\| \frac{1}{N} \sum_{i=1}^N \sigma_i (W\mathbf{z}_i + \mathbf{b}) \right\|_{\ell_T^\infty(\mathbb{R}^{n_v})} \right] + \frac{1}{2\sqrt{N}},$$

because the sigmoid is 1-Lipschitz and $\rho(0) = 0.5$. Now we can apply Lemma B.1 (see Remark B.2) to get that

$$\mathbb{E}_{\boldsymbol{\sigma}} \left[ \sup_{(W,\mathbf{b}) \in \mathcal{W} \times \mathcal{B}} \sup_{\{\mathbf{z}_i\}_{i=1}^N \in Z} \left\| \frac{1}{N} \sum_{i=1}^N \sigma_i (W\mathbf{z}_i + \mathbf{b}) \right\|_{\ell_T^\infty(\mathbb{R}^{n_v})} \right] + \frac{1}{2\sqrt{N}}$$

$$\leq \sup_{W \in \mathcal{W}} \|W\|_{\infty,\infty} \, \mathbb{E}_{\boldsymbol{\sigma}} \left[ \sup_{\{\mathbf{z}_i\}_{i=1}^N \in Z} \left\| \frac{1}{N} \sum_{i=1}^N \sigma_i \mathbf{z_i} \right\|_{\ell_T^\infty(\mathbb{R}^{n_u})} \right] + \frac{1}{\sqrt{N}} \sup_{\mathbf{b} \in \mathcal{B}} \|\mathbf{b}\|_\infty + \frac{1}{2\sqrt{N}}$$

Therefore, the sigmoid MLP layer is $(K_W, K_{\mathbf{b}} + 0.5)$-RC. The restriction of an MLP to the ball $B_{\ell_T^\infty(\mathbb{R}^m)}(r)$ maps to the ball $B_{\ell_T^\infty(\mathbb{R}^{n_v})}(1)$, because of the elementwise sigmoid activation.

**Single layer MLP with ReLU activations.** We repeat the same steps as in the proof up to the first inequality. Here we can apply Equation 4.20 from Ledoux & Talagrand (1991) (this is the same idea as in the proof of Lemma 2 in Golowich et al. (2018)) to get

$$\mathbb{E}_{\boldsymbol{\sigma}} \left[ \sup_{h \in \mathcal{H}} \left\| \frac{1}{N} \sum_{i=1}^N \sigma_i \rho(h(\mathbf{x}_i)) \right\|_\infty \right] \leq 4 \mathbb{E}_{\boldsymbol{\sigma}} \left[ \sup_{h \in \mathcal{H}} \left\| \frac{1}{N} \sum_{i=1}^N \sigma_i h(\mathbf{x}_i) \right\|_\infty \right],$$

where we used that $\rho(x) = ReLU(x)$ is 1-Lipschitz and $\rho(0) = 0$, and the same logic for the alternative definition of the Rademacher complexity (without the absolute value) as in the proof of Proposition 6.2 Hajek & Raginsky (2019). The constant 4 is then obtained by the additional constant factor 2 from Talagrand's lemma. The rest of proof is identical to the sigmoid case.

The restriction of an MLP to the ball $B_{\ell_T^\infty(\mathbb{R}^{n_u})}(r)$ maps to the ball $B_{\ell_T^\infty(\mathbb{R}^{n_v})}(K_W r + K_{\mathbf{b}})$, because the elementwise ReLU does not increase the infinity norm, hence we can apply Lemma B.1 and Remark B.2. Again, for the deep model the result is straightforward from Lemma 5.2 along with Lemma B.1, Remark B.2.

**GLU.** For the ease of notation, assume that $K_{GLU,i} = K_{GLU}$ and let $\mathcal{W} = \{W \in \mathbb{R}^{n_y \times n_u} \mid \|W\|_{\infty,\infty} < K_{GLU}\}$ and let $\mathcal{F}_{\text{GLU}} = \{f_{GLU}$ as in equation 7 $\mid W \in \mathcal{W}\}$. As $\mathcal{F}_i \subseteq \mathcal{F}_{\text{GLU}}$, it is enough to prove the claim of the lemma for $\mathcal{F}_{\text{GLU}}$.

First of all, we show that the function $h : (\mathbb{R}^2, \|\cdot\|_2) \to (\mathbb{R}, |\cdot|)$ defined as $h(\mathbf{x}) = x_1 \cdot \sigma(x_2)$ is $\sqrt{2}(K+1)$-Lipschitz on a bounded domain, where $|x_i| \leq K$ for all $\mathbf{x} \in \mathbb{R}^2$ we consider. We will later specify the value of $K$ in relation to Assumption 4.10. By the sigmoid being 1-Lipschitz, we have

$$|h(\mathbf{x}) - h(\mathbf{y})| = |x_1 \sigma(x_2) - y_1 \sigma(x_2) + y_1 \sigma(x_2) - y_1 \sigma(y_2)| \leq$$
$$|(x_1 - y_1)\sigma(x_2)| + |y_1(\sigma(x_2) - \sigma(y_2))| \leq |x_1 - y_1| + |y_1||x_2 - y_2|$$
$$\leq \sqrt{2}(K+1) \|\mathbf{x} - \mathbf{y}\|_2$$

Second, we recall Corollary 4 in Maurer (2016).

**Theorem B.4** (Maurer (2016)). *Let $\mathcal{X}$ be any set, $(\mathbf{x}_1, \ldots, \mathbf{x}_N) \in \mathcal{X}^N$, let $\widetilde{\mathcal{F}}$ be a set of functions $f : \mathcal{X} \to \ell_T^2(\mathbb{R}^m)$ and let $h : \ell_T^2(\mathbb{R}^m) \to \mathbb{R}$ be an $L$-Lipschitz function. Under $f_k$ denoting the $k$-th component function of $f$ and $\sigma_{ik}$ being a doubly indexed Rademacher variable, we have*

$$\mathbb{E}_{\boldsymbol{\sigma}} \left[ \sup_{f \in \widetilde{\mathcal{F}}} \sum_{i=1}^N \sigma_i h(f(\mathbf{x}_i)) \right] \leq \sqrt{2} L \mathbb{E}_{\boldsymbol{\sigma}} \left[ \sup_{f \in \widetilde{\mathcal{F}}} \sum_{i=1}^N \sum_{k=1}^m \sigma_{ik} f_k(\mathbf{x}_i) \right].$$

We wish to apply Theorem B.4 to GLU layers. For any $Z \subseteq \ell_T^\infty(\mathbb{R}^{n_u})$, by letting $GLU_W(\mathbf{z}) = f_{GLU}(\mathbf{z})$ we have

$$\mathbb{E}_{\boldsymbol{\sigma}} \left[ \sup_{W \in \mathcal{W}} \sup_{\{\mathbf{z}_i\}_{i=1}^N \in Z} \left\| \frac{1}{N} \sum_{i=1}^N \sigma_i GLU_W(\mathbf{z}_i) \right\|_{\ell_T^\infty(\mathbb{R}^{n_u})} \right]$$
$$= \mathbb{E}_{\boldsymbol{\sigma}} \left[ \sup_{W \in \mathcal{W}} \sup_{\{\mathbf{z}_i\}_{i=1}^N \in Z} \sup_{1 \leq k \leq T} \sup_{1 \leq j \leq n_u} \left| \frac{1}{N} \sum_{i=1}^N \sigma_i GLU_W^{(j)}(\mathbf{z}_i)[k] \right| \right].$$

Now this is an alternative version of the Rademacher complexity, where we take the absolute value of the Rademacher average. In order to apply Theorem B.4, we reduce the problem to the usual Rademacher

complexity. In turn, we can apply the last chain of inequalities in the proof of Proposition 6.2 in Hajek & Raginsky (2019). Concretely, by denoting $\mathbf{O} = \{\mathbf{0}\}_{i=1}^{N}$ and noticing that $GLU_W(0) = 0$, we have

$$
\mathbb{E}_{\boldsymbol{\sigma}}\left[\sup_{W \in \mathcal{W}} \sup_{\{\mathbf{z}_i\}_{i=1}^N \in Z} \sup_{1 \leq k \leq T} \sup_{1 \leq j \leq n_u} \left| \frac{1}{N} \sum_{i=1}^{N} \sigma_i GLU_W^{(j)}(\mathbf{z}_i)[k] \right| \right]
$$
$$
\leq 2\mathbb{E}_{\boldsymbol{\sigma}}\left[\sup_{W \in \mathcal{W}} \sup_{\{\mathbf{z}_i\}_{i=1}^N \in Z \cup \{\mathbf{O}\}} \sup_{1 \leq k \leq T} \sup_{1 \leq j \leq n_u} \frac{1}{N} \sum_{i=1}^{N} \sigma_i GLU_W^{(j)}(\mathbf{z}_i)[k] \right].
$$

Let $\mathbf{x}_i = i$, $i = 1, \ldots, N$ and let $\mathcal{H} = \{f_{W,\underline{z},k,j} \mid (W, \underline{z}, k, j) \in \mathcal{W} \times (Z \cup \{0\}) \times [T] \times [n_u]\}$ such that $f_{W,\underline{z},k,j}(\mathbf{x}_i) = \left(GELU(\mathbf{z}_i[k])^{(j)}, \ (W(GELU(\mathbf{z}_i[k])))^{(j)}\right)^T$ for $\underline{z} = \{\mathbf{z}_i\}_{i=1}^N \in Z$. Since $Z \subseteq (B_{\ell_T^\infty(\mathbb{R}^{n_y})}(r))^N$, it follows that for all $\{\mathbf{z}_i\}_{i=1}^N \in Z$ and for all $k \in \mathbb{N}$, the inequality $\|\mathbf{z}_i[k]\|_\infty \leq r$ holds. Hence, it follows that $|GELU(\mathbf{z}_i[k])^{(j)}| < r$, leading to $|W(GELU(\mathbf{z}_i[k]))^{(j)}| < \sup_{W \in \mathcal{W}} \|W\|_{\infty,\infty} \cdot r$. In particular, $GLU_W^{(j)}(\mathbf{z}_i)[k] = h(f_{W,\underline{z},k,j}(\mathbf{x}_i)) = h|_B(f_{W,\underline{z},k,j}(\mathbf{x}_i))$, where $h|_B$ is the restriction of $h$ to $B = \{\mathbf{x} \in \mathbb{R}^2 \mid \|\mathbf{x}\|_\infty < K\}$, with $K = \max\{r, \sup_{W \in \mathcal{W}} \|W\|_{\infty,\infty} r\}$. In particular, $h|_B$ is $\sqrt{2}(K+1)$-Lipschitz.

We are ready to apply Theorem B.4, together with the GLU definition and its $\sqrt{2}(K+1)$-Lipschitzness, we have

$$
2\mathbb{E}_{\boldsymbol{\sigma}}\left[\sup_{W \in \mathcal{W}} \sup_{\{\mathbf{z}_i\}_{i=1}^N \in Z \cup \{\mathbf{O}\}} \sup_{1 \leq k \leq T} \sup_{1 \leq j \leq n_u} \frac{1}{N} \sum_{i=1}^{N} \sigma_i GLU_W^{(j)}(\mathbf{z}_i)[k] \right]
$$
$$
= 2\mathbb{E}_{\boldsymbol{\sigma}}\left[\sup_{f \in \mathcal{H}} \frac{1}{N} \sum_{i=1}^{N} \sigma_i h(f(\mathbf{x}_i)) \right] \leq 4(K+1)\mathbb{E}_{\boldsymbol{\sigma}}\Bigg[\underbrace{\sup_{\{\mathbf{z}_i\}_{i=1}^N \in Z \cup \{\mathbf{O}\}} \frac{1}{N} \sum_{i=1}^{N} \sigma_i GELU(\mathbf{z}_i[k])^{(j)}}_{A}\Bigg]
$$
$$
+ 4(K+1)\mathbb{E}_{\boldsymbol{\sigma}}\Bigg[\underbrace{\sup_{\{\mathbf{z}_i\}_{i=1}^N \in Z \cup \{\mathbf{O}\}} \frac{1}{N} \sum_{i=1}^{N} \sigma_i W(GELU(\mathbf{z}_i))^{(j)}[k]}_{B}\Bigg]
$$

Due to the definition of GELU, its 2-Lipschitzness (Qi et al., 2023) and Theorem 4.12 from Ledoux & Talagrand (1991) we have

$$
A = \mathbb{E}_{\boldsymbol{\sigma}}\left[\sup_{\{\mathbf{z}_i\}_{i=1}^N \in Z \cup \{\mathbf{O}\}} \left\| \frac{1}{N} \sum_{i=1}^{N} \sigma_i GELU(\mathbf{z}_i) \right\|_{\ell_T^\infty(\mathbb{R}^{n_u})} \right] =
$$
$$
\leq 4\mathbb{E}_{\boldsymbol{\sigma}}\left[\sup_{\{\mathbf{z}_i\}_{i=1}^N \in Z \cup \{\mathbf{O}\}} \left\| \frac{1}{N} \sum_{i=1}^{N} \sigma_i \mathbf{z}_i \right\|_{\ell_T^\infty(\mathbb{R}^{n_u})} \right] = 4\mathbb{E}_{\boldsymbol{\sigma}}\left[\sup_{\{\mathbf{z}_i\}_{i=1}^N \in Z} \left\| \frac{1}{N} \sum_{i=1}^{N} \sigma_i \mathbf{z}_i \right\|_{\ell_T^\infty(\mathbb{R}^{n_u})} \right]
$$

and

$$
B = \mathbb{E}_{\boldsymbol{\sigma}}\left[\sup_{W \in \mathcal{W}} \sup_{\{\mathbf{z}_i\}_{i=1}^N \in \{\mathbf{O}\}} \left\| \frac{1}{N} \sum_{i=1}^{N} \sigma_i W(GELU(\mathbf{z}_i)) \right\|_{\ell_T^\infty(\mathbb{R}^{n_u})} \right]
$$
$$
\leq \sup_{W \in \mathcal{W}} \|W\|_\infty \mathbb{E}_{\boldsymbol{\sigma}}\left[\sup_{\{\mathbf{z}_i\}_{i=1}^N \in Z\{\mathbf{O}\}} \left\| \frac{1}{N} \sum_{i=1}^{N} \sigma_i GELU(\mathbf{z}_i) \right\|_{\ell_T^\infty(\mathbb{R}^{n_u})} \right]
$$
$$
\leq 4 \sup_{W \in \mathcal{W}} \|W\|_\infty \mathbb{E}_{\boldsymbol{\sigma}}\left[\sup_{\{\mathbf{z}_i\}_{i=1}^N \in Z} \left\| \frac{1}{N} \sum_{i=1}^{N} \sigma_i \mathbf{z}_i \right\|_{\ell_T^\infty(\mathbb{R}^{n_u})} \right]
$$

Here we used the linearity of $W$ and the exact same calculation as in the proof of Lemma B.1. By combining the inequalities above, it follows that

$$\mathbb{E}_{\boldsymbol{\sigma}}\left[\sup_{W\in\mathcal{W}}\sup_{\{\mathbf{z}_i\}_{i=1}^N\in Z}\sup_{1\leq k\leq T}\sup_{1\leq j\leq n_u}\left|\frac{1}{N}\sum_{i=1}^N\sigma_i GLU_W^{(j)}(\mathbf{z}_i)[k]\right|\right]\leq$$

$$16(K+1)\left(\sup_{W\in\mathcal{W}}\|W\|_{\infty,\infty}+1\right)\mathbb{E}_{\boldsymbol{\sigma}}\left[\sup_{\{\mathbf{z}_i\}_{i=1}^N\in Z}\left\|\frac{1}{N}\sum_{i=1}^N\sigma_i\mathbf{z}_i\right\|_{\ell_T^\infty(\mathbb{R}^{n_u})}\right]$$

Substituting the value of $K$ gives the result.

**SSM block.** By Lemma 5.2 and the proof of this lemma for SSM layers and non-linear layers $\mathcal{F}_i$, we have that the composition of the SSM layer $\mathcal{E}$ and a non-linear layer $\mathcal{F}_i$ is $(\mu_i(rK_1)K_1, c_i(rK_1))$-RC for $i > 1$ and it is $(\mu_i(K_2 r)K_2, c_i(K_2 r))$-RC for $i = 1$ and its elements map a ball of radius $r$ to a ball of radius $\hat{r}_i(K_1 r)$ for $i > 1$ and to ball of radius $\hat{r}_i(K_2 r)$ for $i = 1$. A SSM block is then $(\mu_i(K_{l(i)}r)K_{l(i)} + \alpha_i, c_i(K_{l(i)}r))$-RC, where $K_{l(1)} = K_2$ $K_{l(i)} = K_1$, $i > 1$ because

$$\mathbb{E}_{\boldsymbol{\sigma}}\left[\sup_{g\in\mathcal{F}_i,\Sigma\in\mathcal{E}}\sup_{\{\mathbf{z}_j\}_{j=1}^N\in Z}\left\|\frac{1}{N}\sum_{j=1}^N\sigma_j(g(\mathcal{S}_\Sigma(\mathbf{z}_j))+\alpha\mathbf{z}_j)\right\|_{\ell_T^\infty(\mathbb{R}^{n_u})}\right]\leq$$

$$\mathbb{E}_{\boldsymbol{\sigma}}\left[\sup_{g\in\mathcal{F}_i,\Sigma\in\mathcal{E}}\sup_{\{\mathbf{z}_j\}_{j=1}^N\in Z}\left\|\frac{1}{N}\sum_{j=1}^N\sigma_j g(\mathcal{S}_\Sigma(\mathbf{z}_j))\right\|_{\ell_T^\infty(\mathbb{R}^{n_u})}\right]+\alpha\mathbb{E}_{\boldsymbol{\sigma}}\left[\sup_{\{\mathbf{z}_j\}_{j=1}^N\in Z}\left\|\frac{1}{N}\sum_{j=1}^N\sigma_j\mathbf{z}_j\right\|_{\ell_T^\infty(\mathbb{R}^{n_u})}\right]$$

$$\leq(\mu_i(r)K_{l(i)}+\alpha_i)\mathbb{E}_{\boldsymbol{\sigma}}\left[\sup_{\{\mathbf{z}_j\}_{j=1}^N\in Z}\left\|\frac{1}{N}\sum_{j=1}^N\sigma_j\mathbf{z}_j\right\|_{\ell_T^\infty(\mathbb{R}^{n_u})}\right]+\frac{c_i(r)}{\sqrt{N}}$$

$\square$

*Proof of Lemma 5.3.* By definition

$$\left\|\frac{1}{N}\sum_{i=1}^N\sigma_i\mathbf{u}_i\right\|_{\ell_T^2(\mathbb{R}^{n_{in}})}=\sqrt{\sum_{k=1}^T\left\|\frac{1}{N}\sum_{i=1}^N\sigma_i\mathbf{u}_i[k]\right\|_2^2}$$

$$=\sqrt{\sum_{k=1}^T\left\langle\frac{1}{N}\sum_{i=1}^N\sigma_i\mathbf{u}_i[k],\frac{1}{N}\sum_{j=1}^N\sigma_j\mathbf{u}_j[k]\right\rangle_{\mathbb{R}^{n_{in}}}}$$

$$=\sqrt{\sum_{k=1}^T\frac{1}{N^2}\sum_{i=1}^N\sum_{j=1}^N\sigma_i\sigma_j\langle\mathbf{u}_i[k],\mathbf{u}_j[k]\rangle_{\mathbb{R}^{n_{in}}}}$$

where $\langle\cdot,\cdot\rangle_{\mathbb{R}^{n_{in}}}$ denotes the standard scalar product in $\mathbb{R}^{n_{in}}$. Therefore

$$\mathbb{E}_{\boldsymbol{\sigma}}\left[\left\|\frac{1}{N}\sum_{i=1}^N\sigma_i\mathbf{u}_i\right\|_{\ell_T^2(\mathbb{R}^{n_{in}})}\right]=\mathbb{E}_{\boldsymbol{\sigma}}\left[\sqrt{\sum_{k=1}^T\frac{1}{N^2}\sum_{i=1}^N\sum_{j=1}^N\sigma_i\sigma_j\langle\mathbf{u}_i[k],\mathbf{u}_j[k]\rangle_{\mathbb{R}^{n_{in}}}}\right]$$

$$\leq\sqrt{\mathbb{E}_{\boldsymbol{\sigma}}\left[\sum_{k=1}^T\frac{1}{N^2}\sum_{i=1}^N\sum_{j=1}^N\sigma_i\sigma_j\langle\mathbf{u}_i[k],\mathbf{u}_j[k]\rangle_{\mathbb{R}^{n_{in}}}\right]}$$

$$= \sqrt{\sum_{k=1}^{T} \frac{1}{N^2} \sum_{i=1}^{N} \sum_{j=1}^{N} \mathbb{E}_{\boldsymbol{\sigma}} \left[\sigma_i \sigma_j\right] \langle \mathbf{u}_i[k], \mathbf{u}_j[k] \rangle_{\mathbb{R}^{n_{\text{in}}}}}$$

$$= \sqrt{\sum_{k=1}^{T} \frac{1}{N^2} \sum_{i=1}^{N} \mathbb{E}_{\boldsymbol{\sigma}} \left[\sigma_i^2\right] \langle \mathbf{u}_i[k], \mathbf{u}_i[k] \rangle_{\mathbb{R}^{n_{\text{in}}}}}$$

$$= \sqrt{\frac{1}{N^2} \sum_{i=1}^{N} \sum_{k=1}^{T} \|\mathbf{u}_i[k]\|_2^2} = \sqrt{\frac{1}{N^2} \sum_{i=1}^{N} \|\mathbf{u}_i\|_{\ell_T^2(\mathbb{R}^{n_{\text{in}}})}^2} \leq \sqrt{\frac{1}{N^2} N K_{\mathbf{u}}^2} \leq \frac{K_{\mathbf{u}}}{\sqrt{N}}$$

$\square$

*Proof of Theorem 5.5.* From Lemma 5.4 it follows that all maps constituting a model $f \in \mathcal{F}$ come from families of maps which are $(\mu, c)$-RC for suitable constants $\mu, c$, and map any ball of radius $r$ to a ball of radius $\hat{r}(r)$. Let us consider the deep SSM model given by equation 9, which is a composite of mappings as

$$B_{\ell_T^2(\mathbb{R}^{n_{\text{in}}})}(K_{\mathbf{u}}) \xrightarrow{\text{Encoder}} B_{\ell_T^2(\mathbb{R}^{n_u})}(\underbrace{K_{\mathbf{u}} K_{\text{Enc}}}_{r_1}) \xrightarrow{\text{B}_1} B_{\ell_T^{\infty}(\mathbb{R}^{n_u})}(r_2) \xrightarrow{\text{B}_2} \dots \xrightarrow{\text{B}_L}$$

$$B_{\ell_T^{\infty}(\mathbb{R}^{n_u})}(r_{L+1}) \xrightarrow{\text{Pooling}} B_{(\mathbb{R}^{n_u}, \|\cdot\|_{\infty})}(r_{L+1}) \xrightarrow{\text{Decoder}} B_{(\mathbb{R}, |\cdot|)}(K_{\text{Dec}} r_{L+1}),$$

where the constants $r_i$, $i \in [L+1]$ are as in equation 13, due to repeated application of Lemma 5.4 and the expressions in Table 3.

Note that the first SSM block is considered as a map $B_{\ell_T^2(\mathbb{R}^{n_u})}(K_{\text{Enc}} K_{\mathbf{u}}) \to B_{\ell_T^{\infty}(\mathbb{R}^{n_u})}(r_2)$, while for $i > 1$ the SSM layer in the $i$th SSM block is considered as a map $B_{\ell_T^{\infty}(\mathbb{R}^{n_u})}(r_i) \to B_{\ell_T^{\infty}(\mathbb{R}^{n_u})}(r_{i+1})$. This is needed, because the encoder is constant in time, therefore the Composition Lemma wouldn't be able to carry the $\ell_T^2$ norm of the input through the chain of estimation along the entire model. This is one of the key technical points which makes it possible to establish a time independent bound.

Next, we wish to apply Theorem 4.2 to the set of deep SSM models $\mathcal{F}$. Let us fix a random sample $S = \{\mathbf{u}_1, \dots, \mathbf{u}_N\} \subset \left(\ell_T^2(\mathbb{R}^{n_{\text{in}}})\right)^N$. As the loss function is Lipschitz according to Assumption 4.10, we have that for any $f \in \mathcal{F}$

$$|\ell(f(\mathbf{u}), y)| \leq 2L_l \max\{f(\mathbf{u}), y\} \leq 2L_l \max\{K_{\text{Dec}} r_{L+1}, K_y\},$$

thus $K_l \leq 2L_l \max\{K_{\text{Dec}} r_{L+1}, K_y\}$. Again by the Lipschitzness of the loss and the Contraction Lemma (Shalev-Shwartz & Ben-David, 2014, Lemma 26.9) we have

$$R_S(L_0) \leq L_l \cdot R_S(\mathcal{F}),$$

and recall that $R_S(\mathcal{F}) = R(\{(f(\mathbf{u}_1), \dots, f(\mathbf{u}_N))^T \mid f \in \mathcal{F}\})$. It is enough to bound the Rademacher complexity $R_S(\mathcal{F})$ of $\mathcal{F}$ to conclude the proof. By applying Lemma 5.4 to every layer of $\mathcal{F}$ and using Lemma 5.2, it follows that the family $\mathcal{F}|_{X_1}$ of restriction of the elements $\mathcal{F}$ to $X_1 = B_{\ell_T^2(\mathbb{R}^{n_{\text{in}}})}(K_{\mathbf{u}})$ is a family of maps from $X_1$ to $X_2 = (\mathbb{R}, |\cdot|)$ which is $(\mu, c)$-RC, where $\mu, c$ are as in equation 14. Next, we state a lemma before we finish the proof.

**Lemma B.5.** *Let $\mathcal{F}$ be a set of functions between $X_1 = B_{\ell_T^2(\mathbb{R}^{n_{\text{in}}})}(K_{\mathbf{u}})$ and $X_2 = (\mathbb{R}, |\cdot|)$ that is $(\mu, c)$-RC. The Rademacher complexity of $\mathcal{F}$ w.r.t. some dataset $S$ for which Assumption 4.10 holds, admits the following inequality.*

$$R_S(\mathcal{F}) \leq \frac{\mu K_{\mathbf{u}} + c}{\sqrt{N}}.$$

*Proof.*

$$R_S(\mathcal{F}) = R(\{(f(\mathbf{u}_1), \dots, f(\mathbf{u}_N))^T \mid f \in \mathcal{F}\}) = \mathbb{E}_\sigma \left[\sup_{f \in \mathcal{F}} \frac{1}{N} \sum_{i=1}^{N} \sigma_i f(\mathbf{u}_i)\right]$$

$$\leq \mathbb{E}_\sigma\left[\sup_{f\in\mathcal{F}}\left|\frac{1}{N}\sum_{i=1}^N \sigma_i f(\mathbf{u}_i)\right|\right] \leq \mu\mathbb{E}_\sigma\left[\left\|\frac{1}{N}\sum_{i=1}^N \sigma_i \mathbf{u}_i\right\|_{\ell_T^2(\mathbb{R}^{n_{\text{in}}})}\right] + \frac{c}{\sqrt{N}}$$

By applying Lemma 5.3, it follows

$$R_S(\mathcal{F}) \leq \frac{\mu K_{\mathbf{u}} + c}{\sqrt{N}}$$

□

The Theorem is then a direct corollary of Lemma B.5. □

