# OpenReview forum: "Length independent generalization bounds for deep SSM architectures via Rademacher contraction and stability constraints"
_TMLR — Accepted by TMLR_

### Review · Reviewer_NLzu · 2025-08-06

**Summary Of Contributions:**

The paper derives an upper bound on the Rademacher complexity of a set of deep SSMs; combined with existing theorems this bound leads to a PAC generalisation bound. The proof relies on a recursive contraction argument that reduces the rademacher complexity of the SSM to the rademacher complexity of the inputs by propagating back through the layers. Empirically, the bound is shown to be non-trivial.

**Audience:**

Yes

**Broader Impact Concerns:**

None.

**Claims And Evidence:**

Yes

**Requested Changes:**

See question above.

**Strengths And Weaknesses:**

Strengths

- First PAC bound for deep SSM.
- The argument is clear and easy to follow

Question

- Could you elaborate on the following? “The bound remains independent of the sequence length, unlike the other sequential model bounds that diverge with T”. Is this a consequence of the bound on the inputs of sequence length T, K_u?

I am not an expert on SSM/PAC bounds so my review is of lower confidence.

---

> ### Author Response · Authors · 2025-08-16
> **Response to Reviewer NLzu**
>
> We thank the Reviewer for their time and effort spent on the review. The newly added parts of the text in the revised manuscript are written in red.
>
> The independence of $T$ arises from the stability of the SSM blocks and the assumption that $K_u$ bounds the $\ell_2$ norm of input sequences of length $T$. This assumption is necessary; however, without stability, it is not sufficient to ensure the bound is independent of $T$. A more detailed explanation is provided below.
>
> The proof relies on separate estimation of the constants $\mu$ and $c$ for each component of each layer in a deep SSM model, and then the application of the Composition Lemma (Lemma 5.2 in the paper). Therefore, if the bound depended on the sequence length $T$, the only way it could happen is that some of the terms $\mu$ and $c$ depend on $T$. The only components that are time-mixing are the LTI (SSM) layers, thus no other component could cause a time-dependent bound.
>
> From the proof of Lemma 5.4, more precisely from the proof of Lemma B.1, it turns out that for the LTI (SSM) layers, we have $c = 0$ due to its linearity in the input sequence.
>
> As for $\mu$, it turns out that the linearity of the LTI layer together with the stability assumption (5. assumption in Assumption 4.10) imply that the constant $\mu$ is independent of $T$.
>
> Using the Composition Lemma (Lemma 5.2 in the paper) repeatedly for all layers, it follows that there exist suitable constants $\tilde{\mu}$ and $\tilde{c}$, which do not depend on $T$, such that we can bound the Rademacher complexity of the model by $\tilde{\mu} \cdot \mbox{Rademacher complexity of the inputs} + \tilde{c}$.
> In turn, Rademacher compexity of the inputs can be bounded (see page 383 of Shalev-Shwartz et. al 2014) using the bound $K_u$ on the $\ell_2$ norm of input sequences of length $T$ introduced in Assumption 4.10. The term $K_u$ is introduced to denote this bound on the input norms. Note that the bound on the input norms $K_u$ is assumed to be independent on $T$ (which we consider a realistic assumption).
>
> We remark that our bound would also hold if one only had a time-dependent bound on the input norms, however the bound itself would be dependent on $T$ in this case. We also remark that the stability property implies that we have a time-independent bound on the operator norm of the LTI (SSM) layer of the model. If instead one only had a time-dependent upper bound on this operator norm, our bound would again hold and again it would be time-dependent.
>
> A brief version of the above explanation is part of the newly added Remark 5.7 in Section 5 of the revised manuscript.

---

### Review · Reviewer_rrwk · 2025-08-10

**Summary Of Contributions:**

The paper analyzes deep state-space architectures under standard stability assumptions and examines the Rademacher complexity of these networks to derive generalization bounds that do not scale with input length.

**Audience:**

Yes

**Broader Impact Concerns:**

None.

**Claims And Evidence:**

Yes

**Requested Changes:**

Since, in my opnion, the paper is well structured and well motivated, I think it would most benefit from clarifying the novelty of its proof techniques: explain why prior methods don’t directly apply here and pinpoint where the proof introduces new ideas. If there is no genuine novelty, state this explicitly (I’m not sure that alone should warrant rejection.)

**Strengths And Weaknesses:**

## Strengths

1. The paper is well organized and clearly written. The “informal statement of results” section motivates the work and summarizes the contributions, and the assumptions section helpfully frames the (relatively small) limits of theory. Overall I think the structure is great.
2. The problem is well motivated: SSMs are governed by their impulse responses, whose norms are bounded by something proportional to the system’s norm, so it’s natural that generalization need not depend on sequence length; the paper presents itself as the first to make this precise.
3. The assumptions strike me as very reasonable, which is refreshing for modern NN theory.
4. I haven’t checked every detail, but the proofs appear sound.

## Weaknesses

1. I’m not a Rademacher-complexity expert, but compared with prior work the technical development doesn’t feel especially novel and seems to reuse familiar arguments. The proposed “$(\mu,c)$-Rademacher Contraction” looks close to the standard contraction lemma (The specific case of $c=0$). I’m not convinced the level of technical novelty clears the bar set by the related literature.
2. The length-independent generalization bounds are not surprising to me (as explained in strength number 2), which is not true in setting investigated in prior works, such as RNNs, whose behavior can intuitively vary more with length). While the problem is well motivated, my intuition is that the result mainly repackages established techniques, though I’m not expert enough to be certain.

---

> ### Author Response · Authors · 2025-08-16
> **Response to Reviewer rrwk**
>
> We thank the Reviewer for their time and effort spent on the review. We reply to the weaknesses and questions in the same order the Reviewer mentioned them. The newly added parts of the text in the revised manuscript are written in red.
>
> __1. Novelty of the concept of Rademacher contraction.__
>
> To the best of our knowledge, the concept of Rademacher contraction (RC) is new. While the classical contraction lemma can be seen as a special case of RC, the latter applies more broadly - and this generality is crucial to our proof. As noted in the paper, similar techniques appear in derivations of Rademacher complexity bounds for deep neural networks (see Golowich et al., 2018). Using RC makes these connections more explicit.
>
> More precisely, the standard contraction lemma can only be straightforwardly applied when the nonlinearity following the parametrized previous layer is not parametrized, i.e. it is known and not learned. For instance, an MLP layer is a parametrized, linear component followed by a fixed activation function. In this case, Talagrand's lemma has been exploited in previous works (e.g. Golowich et al., 2018).
>
> This case does not cover SSM layers or GLU layers, so the contraction lemma cannot be applied directly to these blocks. Instead, we used Rademacher Contractions (RC). For SSM blocks, we developed a new proof; for GLU blocks, we relied on Maurer (2016) which extends the contraction lemma to vector-valued functions, along with several additional steps. Even for MLP blocks, we had to use results from Golowich et al. (2018), which do follow from the contraction lemma but only through a nontrivial derivation.
>
> Beyond addressing the specific technical challenges in this paper,  we hope that RC will prove useful for other deep models.  Since RC is defined for general models acting on Banach spaces  and is closed under composition,  it enables recursive bounds on the Rademacher complexity of an $n$-layer model  using bounds for $(n-1)$-layer models,  which might particularly be useful for analyzing more general deep architectures.
>
> __2. "Repackaging established techniques".__ The Reviewer is right in the sense that many of the techniques used in the proofs (PAC bounds based on Rademacher compexity, etc.)  are well-known.
> However, there are two tools in our work which are new:
> the use of Rademacher Contractions (RC) and the use of
> the $H_2$ and $\ell_1$ norms of SSMs from control theory.
> Both techniques may have applications beyond deep SSMs.
> As noted, RC can also be applied to deep models other than SSMs.
> Moreover, linking classical control-theoretic tools
> (stability and system norms) with machine-learning tools
> (Rademacher complexity and PAC generalization bounds)
> could prove valuable for analyzing other dynamical models as well.
>
> __3. Summary of the novelty in the proof techniques.__
>
> - The introduced $(\mu, c)$-RC inequality can be seen as a generalization of Talagrand's lemma and similar techniques, and can be applied to time-mixing model components which act on Banach-spaces
> of sequences and which contain components with parametrized nonlinearities.
> - RC is general enough to cover time-mixing layers and time-independent nonlinearities (such as MLPs or even GLU layers) and it is invariant under function composition.
> - The use of system norms. RC allows us to make a formal connection between the system norms and the degree of stability of the time-mixing components, and the generalization error of the model. In broader sense, it connects the field of control to the field of machine learning.
>
> __4. Summary of why prior methods for RNNs/SSMs don't work.__
>
> - Proofs on simple RNNs don't automatically apply as deep SSMs and simple RNNs are fundamentally different structures, see Remark 4.9.
> - Proofs for single layer, linear SSMs (LTI systems) don't apply as we consider deep structures with nonlinearities.
> - Proofs based on Talagrand's lemma don't automatically apply as in the general case, the considered SSM blocks are made of a time mixing layer followed by a parametrized nonlinearity instead of a fixed activation.
> - Proof techniques based on upper bounding the Rademacher complexity by upper bounding some covering numbers don't apply as it is not clear how to bound covering numbers of deep SSMs, even if the covering
> numbers of the individual layers are known, as the behavior of covering numbers under model composition is
> not straightforward.
> - Many existing proof ideas would result on a time-dependent bound and require stronger assumptions, such as bounded parameter norms in contrast to bounded system norms. This is a weaker assumption as it is possible to define SSM layers with arbitrarily large parameter norm, but which are stable
> with uniformly bounded $H_2$ or $\ell_1$ system norms.
>
> We incorporated the above discussion in the revised manuscript by extending the paragraph "__Rademacher Contractions.__" in Section I. and also by adding the new Remarks 5.6 and 5.7 in Section 5.

---

### Review · Reviewer_nWKz · 2025-08-11

**Summary Of Contributions:**

This paper focuses on deep SSMs and provide a PAC bound on their generalization error, i.e., the difference between the true loss and the empirical loss. The result relates stability of the SSM blocks with performance, and does not depend on the length of the input sequences.

**Audience:**

Yes

**Broader Impact Concerns:**

I don't foresee relevant ethical implications of the work.

**Claims And Evidence:**

Yes

**Requested Changes:**

I believe the paper should be made clearer, and also targeting a broader audience, following for example my notes above.

**Strengths And Weaknesses:**

This paper provides a novel and significant result on the theoretical properties of deep SSMs, specifically their generalization error. The result appears to be technically challenging to obtain. Additionally, the result provides a relationship between stability of the SSM blocks and the overall performance. Finally, the notion of Rademacher contraction that is introduces is interesting and could be of use in other contexts.

On the downside, I believe the paper would benefit from being clearer in some parts, and aim for a broader audience in others. More specifically:
- The abstract says deep SSMs are good for long-range sequences, but to do which tasks with those sequences?
- The abstract says the paper provides a PAC bound, but a PAC bound on which property?
- The introduction says stability is is key, but none of the contributions highlighted mention it;
- "LTIs" are mentioned throughout without explanation; also MLP and GLU should be clarified, even though those two are more broadly understandable;
- Almost all references should be cited between brackets, because it becomes hard to distinguish what are citations and what are the words in the sentence;
- Can we obtain an intuitive explanation of Rademacher complexity along with the formal definition?
- Some notation is not defined ("n_u", "n_y", "B_L", "\sigma_1", etc.);
- As far as I understand, it is stated that Schur matrices are equivalent to having the eigenvalues inside the unit disk. I don't see how this is true because for example 2*I (I is the identity) is clearly Schur (since 2*I = I*(2*I)*I^{-1}) and the eigenvalues are clearly not inside the disk;
- The relation between the experimental results on the plots and the theoretical result is not clear to me. I see there is an effort to make this connection in the text, but I would appreciate if it could be made clearer, both in the text and in the figure captions;
- What are Markov parameters?

---

> ### Author Response · Authors · 2025-08-16
> **Response to Reviewer nWKz**
>
> We thank the Reviewer for their time and effort spent on the review. We extended the manuscript with additional explanations along the points raised by the Reviewer. For the sake of completeness, we also explicitly answer these points below, in the same order as the Reviewer raised them. The newly added parts of the text in the revised manuscript are written in red.
>
> - We modified the abstract: "... achieving excellent performance on learning
> representations of long-range sequences".
> - We modified the abstract: "... a PAC bound on the
> generalization error of non-selective architectures with stable SSM blocks, that...".
> - We modified the contribution highlights in the Introduction in the revised manuscript to reflect the role of stability in the contributions of the paper.
> - The Reviewer is right, the term LTI system (Linear Time-Invariant system) is left undefined, we fixed it in the revision. The terms MLP and GLU are defined in Definition 4.4 and 4.5, however we added additional references to these definitions for the first time they are referenced in the text.
> - We changed the references according to the directives in the TMLR template file, i.e. according to "When the authors or the publication are included in the sentence, the citation should not be in parenthesis, using \citet{} (as in “See Hinton et al. (2006) for more information.”). Otherwise, the citation should be in parenthesis using \citep{} (as in “Deep
> learning shows promise to make progress towards AI (Bengio \& LeCun, 2007).”).".
> - We added an intuitive explanation of Rademacher complexity after its definition in the revised manuscript.
> - We tried our best in the revision to make sure that all variables are explicitly defined.
>  - In the control theory literature, a matrix $A$ is defined to be Schur, if all of its eigenvalues are inside the complex unit disk. We now recognize that within some other fields, a matrix being Schur can have different definitions which are not equivalent to the one we used. We explicitly phrased the definition in the revised manuscript. The matrix $2I$ is not Schur according to the definition we use in the paper, but it might be Schur according to the definition used in other domains.
> - We extended the explanation of the figures. __Figure 2__ is a standard figure related to PAC bounds and it shows two things. First, it shows that the PAC bound of this paper is $O(1 / \sqrt{N})$, and it indeed behaves as predicted by Theorem 5.5. More precisely, let  $\mathcal{L}(f^N)$ be the true loss of the model $f^N$ learned from $N$ data points, and let $r(N,\delta)=\mathcal{L}^S_{emp}(f^N) + \frac{K_{\mathcal{F}} + K_l \sqrt{2 \log(\frac{4}{\delta})}}{\sqrt{N}}$ be  the PAC bound of Theorem 5.5 of the paper. By Theorem 5.5 with probability at least $1-\delta$ on the choice of the training data $S$, $\mathcal{L}(f^N) < r(N, \delta)$.
>     Then Figure 2 shows that for the specific choice of the training dataset it holds that
>     $\mathcal{L}(f^N) \le r(N,\delta)$, i.e., $r(N,\delta)$ is indeed an upper bound on the true loss. It also shows that the bound $r(N,\delta)$ converges to the true loss $\mathcal{L}(f^N)$ at rate $O(\frac{1}{\sqrt{N}})$.
>     Second, it shows that there are scenarios where the bound is non-vacuous, i.e. the bound is meaningful. Namely,
>     we show that for the example at hand, there exist integers $N_1, N_2 \in \mathbb{N}$ such that $\mathcal{L}(f^{N_1}) > r(N_2,\delta)$. This means that the bound $r(N,\delta)$ is not a trivial one, it is not larger for any $N$ than any possible true loss
> $\mathcal{L}(f)$ for any model $f$, i.e., it excludes situations of the type where the true loss for any model is say at most $1$ and the bound is at least $1$.
>
> __Figure 3__ illustrates the behavior of the bound during model training. We empirically found that in this scenario, the dynamics of the bound correlates with the model's generalization performance in accuracy. Namely, the bound stays low up to the point where the accuracy starts to decrease, where the bound starts to grow rapidly. This suggests that adding the bound, or some parts of it, to the loss function as a regularization term during training could lead to faster or more stable models. Again, this is a separate research topic on its own and is out of the scope of the paper.
>
>  - We explicitly defined Markov-parameters (Markov parameters are the members of the set {$D$} $\cup${ $CA^kB$ | $k \in \mathbb{N}$}  and they are important for LTI systems as they uniquely define their input-output map. More precisely, the output sequence $\mathbf{y}$ generated by an LTI system as a response to an input sequence $\mathbf{u}$ satisfies $\mathbf{y}[k] = D\mathbf{u}[k] + \sum\limits_{i=1}^kCA^{i-1}B \mathbf{u} [k-i]$.

---

### Author Response · Authors · 2025-10-20
**Camera ready version uploaded**

We are grateful to the Editor and the Reviewers for their helpful feedback and efforts throughout the process. The camera ready version has been uploaded.

---

### Decision · Action_Editor_934E · 2025-09-20

**Recommendation:** Accept as is

**Audience:**

Yes

**Audience Explanation:**

Generalization of SSM architectures is definitely of interest to a large portion of the community, including both theoreticians and practitioners.

**Claims And Evidence:**

Yes

**Claims Explanation:**

The authors present a theoretical analysis of deep SSMs, deriving PAC generalization bounds that are independent of input sequence length. The arguments are grounded in a Rademacher framework and stability constraints, both motivated and technically sound. The proofs are detailed, and supported by clarifying revisions from the authors. Empirical illustrations further demonstrate the theory. Overall, the paper’s claims are well supported by a combination of mathematical derivations and illustrative experiments